# Activator-blocker model of transcriptional regulation by pioneer-like factors

Aileen Julia Riesle [1,8,9], Meijiang Gao[1,2,9], Marcus Rosenblatt [3,4,9], Jacques Hermes [3,4,9], Helge Hass[3,4], Anna Gebhard[1], Marina Veil[1], Björn Grüning[5,6], Jens Timmer [2,3,4] ✉ & Daria Onichtchouk [1,2,7] ✉

Zygotic genome activation (ZGA) in the development of flies, fish, frogs and mammals depends on pioneer-like transcription factors (TFs). Those TFs create open chromatin regions, promote histone acetylation on enhancers, and activate transcription. Here, we use the panel of single, double and triple mutants for zebrafish genome activators Pou5f3, Sox19b and Nanog, multi-omics and mathematical modeling to investigate the combinatorial mechanisms of genome activation. We show that Pou5f3 and Nanog act differently on synergistic and antagonistic enhancer types. Pou5f3 and Nanog both bind as pioneer-like TFs on synergistic enhancers, promote histone acetylation and activate transcription. Antagonistic enhancers are activated by binding of one of these factors. The other TF binds as non-pioneer-like TF, competes with the activator and blocks all its effects, partially or completely. This activator-blocker mechanism mutually restricts widespread transcriptional activation by Pou5f3 and Nanog and prevents premature expression of late developmental regulators in the early embryo.

Awakening of zygotic transcription during maternal-to-zygotic transition is a universal feature of multicellular organisms. The major wave of Zygotic Genome Activation (ZGA) is driven by sequence-specific transcription factors (TFs), different across species[1,2]. Nucleosomes protect enhancers from binding of most TFs and are the main obstacles for transcriptional activation. The small group of pioneer-like TFs directly bind to nucleosomes and create open, nucleosome-free chromatin regions on enhancers[3]. Zygotic genome activators act as pioneer-like factors, by opening chromatin, initiating enhancer activation, and widespread zygotic gene expression[4,5]. Direct nucleosome binding was shown for some of the genome activators[6,7].

In animals, gene products synthetized at ZGA enable gastrulation and subdivide the embryo into the three germ layers, ectoderm, mesoderm and endoderm. The regulators of later developmental programs, organogenesis and cell lineage specification, are kept silent at ZGA. They will be synthesized shortly before the beginning of appropriate developmental stages[8,9]. The expression waves of transcripts encoding cohorts of transcriptional regulators follow each other in precise order in the developmental time course[10]. Molecular mechanisms which keep lineage specifying genes silent at ZGA are unknown.

Maternal transcription factors Pou5f3, Sox19b and Nanog (PSN) together activate zygotic gene expression in zebrafish[11-13]. Sox19b is the only maternal member of the SoxB1 family. Shortly after ZGA, early zygotic SoxB1 factors Sox3, Sox19a and Sox2 are expressed and act redundantly with Sox19b[14]. PSN, as well as their mammalian

[1]Department of Developmental Biology, Albert-Ludwigs-University of Freiburg, 79104 Freiburg, Germany. [2]Signalling Research centers BIOSS and CIBSS, 79104 Freiburg, Germany. [3]Institute of Physics, Albert-Ludwigs-University of Freiburg, 79104 Freiburg, Germany. [4]Freiburg Center for Data Analysis and Modelling (FDM), 79104 Freiburg, Germany. [5]Department of Computer Science, University of Freiburg, 79110 Freiburg, Germany. [6]Center for Biological Systems Analysis (ZBSA), University of Freiburg, 79104 Freiburg, Germany. [7]Institute of Developmental Biology RAS, 119991 Moscow, Russia. [8]Present address: Epigenetics and Neurobiology Unit, European Molecular Biology Laboratory, EMBL Rome, Adriano Buzzati-Traverso Campus, Via Ramarini 32, 00015 Monterotondo, RM, Italy. [9]These authors contributed equally: Aileen Julia Riesle, Meijiang Gao, Marcus Rosenblatt, Jacques Hermes. ✉e-mail: jeti@fdm.uni-freiburg.de; daria.onichtchouk@biologie.uni-freiburg.de

counterparts, frequently colocalize on the same genomic sites; therefore, they initially were thought to cooperate[12,15]. However, recent studies in mouse and fish revealed that the factors may act alone, additively or interchangeably[5,16,17]. PSN open chromatin on more than half of active enhancers at ZGA, but are necessary for expression of only a fraction of early zygotic genes[5]. It is unclear how widespread pioneer-like activity of PSN on enhancers relates to early zygotic transcription.

The purpose of this study was to understand the mechanisms of combined Pou5f3, Sox19b and Nanog action, and the connection between PSN pioneer-like activity, enhancer activation and selection of the early zygotic gene expression repertoire. We found that Pou5f3 and Nanog in combination not only activate part of zygotic transcripts,

but reciprocally restrict or block transcriptional activation by each other on hundreds of developmental enhancers. Further, competition between Pou5f3 and Nanog on common genomic sites establishes the order of gene expression after ZGA.

## Results

### Study design and overview of the ATAC-seq data set

For multi-omic assays, we used the panel of eight zebrafish genotypes: zebrafish wild-type embryos (WT), maternal-zygotic null-mutants for Sox19b (S, MZsox19b[17]), Pou5f3 (P, MZspg[18]), and Nanog (N, MZnanog[19]), double mutants MZps, MZpn, MZsn, and the triple mutant MZtriple. Sox19b was dispensable for normal development, the absence of Pou5f3, Pou5f3/Sox19b or Nanog resulted in abnormal

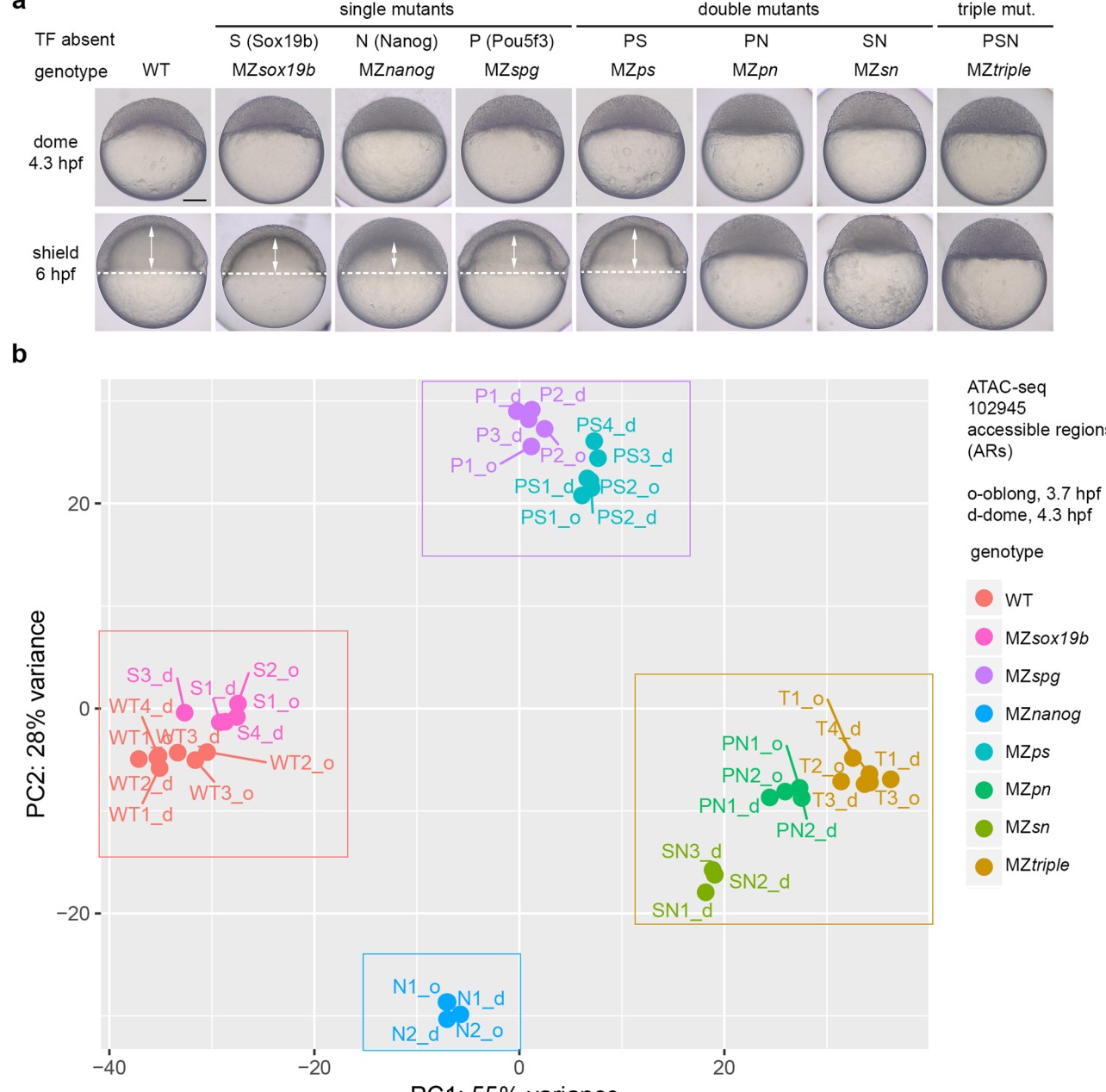

**Fig. 1 | Single and combined mutants by zebrafish zygotic genome activators used in this study. a** Mutant phenotypes at blastula (dome) and midgastrula (shield) stages. White dotted line shows epiboly border; double arrows show the internalized yolk. Scale bar=100 μm. **b** Principal Component Analysis of ATAC-seq data in all mutants, on the genomic regions accessible in the wild-type (ARs). The data in eight genotypes clustered into four groups (boxed). Source data are provided as a Source Data file 1.

gastrulation as previously reported[17–19]. At the absence of Pou5f3/Nanog, Sox19b/Nanog or all three factors zygotic development was arrested (Fig. 1a).

To assess individual and combined effects of Pou5f3, Sox19b, and Nanog on enhancer regulation and transcription, we performed ATAC-seq and time-resolved RNA-seq profiling in all genotypes. We then aimed to derive enhancer groups differentially regulated by PSN using genomic data (Source Data file 1), and differentially regulated target gene groups using transcriptome (Source Data file 2). Finally, for cross-validation, we put together the independently obtained results of the genome and transcriptome parts (Source Data file 3). The outline of data analysis is shown in Fig. S1.

We started the genomic analysis by estimating the differences in the chromatin accessibility between the genotypes, on 102945 Accessible Regions (ARs) in the wild-type. We made three initial observations using Principal Component Analysis (PCA, Fig. 1b). First, chromatin accessibility profiles in MZ*sox19b* mutants were close to the wild-type. Sox19b is the major SoxB1 factor expressed at 3.7 and 4.3 hpf, we therefore assumed that the Sox19b factors acts as a cofactor or redundantly with Pou5f3 or Nanog. Second, accessibility profiles of MZ*sn*, MZ*pn*, and MZ*triple* mutants clustered together in PCA space, corresponding to the most severe phenotypes of these mutants. Third, accessibility profiles of Pou5f3 and Nanog single mutants were quite different, suggesting that these TFs change chromatin accessibility on different regions, or in the opposite directions.

### Analysis of MZ*triple* mutant

To tease apart the apparently complex regulation of chromatin accessibility by Pou5f3, SoxB1 and Nanog, we first characterized the effects of their combined activity on chromatin using MZ*triple* mutant. We divided all ARs to three groups: "down", where the chromatin accessibility was reduced in MZ*triple* compared to the wild-type (31% of all ARs), "up", where the chromatin accessibility was increased (15% of all ARs), and "same" –unchanged (Fig. 2a). Sequence-specific binding motifs for all three factors were enriched in "down" regions, while "up" regions had the highest content of G and C nucleotides and of GC-rich TF-binding motifs (Fig. 2b, c). The genes closest to "down" and "up" ARs were both enriched in transcriptional regulatory functions among the others (Fig. 2d).

Next, we analyzed the time-resolved transcriptome in MZ*triple* mutant compared to the wild-type, using our previously developed R package RNA-sense (www.bioconductor.org/packages/release/bioc/html/RNAsense.html[17]). We selected 4777 transcripts zygotically expressed in the wild-type or in MZ*triple*, and split them into three groups (Fig. 2e): "DOWN" (downregulated compared to the wild-type or not expressed in MZ*triple*), "SAME" (unchanged), and "UP" (upregulated in MZ*triple* compared to the wild-type, or expressed only in MZ*triple*).

Out of 791 "UP" group transcripts, 67% were zygotically expressed only in MZ*triple* (i.e. *hoxc8a*, *nr2f2*). We wondered if lineage-specifying regulatory genes were prematurely expressed MZ*triple*, as we observed previously in MZ*spg* and MZ*ps* mutants[17,20]. To test that, we used published data[10] to calculate the maximal expression time during normal zebrafish development for each zygotic gene. Indeed, the median expression time in the "UP" group was 24 hpf, versus 8 hpf in the groups "DOWN" and "SAME" (Fig. 2f). Similar analysis for other mutants revealed that except for MZ*sox19b*, all of them prematurely expressed diverse sets of late regulatory genes (Fig. S2).

Finally, we checked if the changes in chromatin accessibility on the putative regulatory regions of zygotic genes correlated with the changes in their expression in MZ*triple*. This was indeed the case: "down" ARs were enriched around the promoters of "DOWN" genes, and "up" ARs around "UP" genes in MZ*triple* (Fig. 2g).

Taken together, our analysis confirmed previous findings that Pou5f3, Sox19b and Nanog initiate chromatin accessibility on the

enhancers of early zygotic genes[5,21,22]. The increase in transcription in MZ*triple* mutants was unexpected. We concluded that additional activators were present at ZGA. These unknown activators could bind to GC-rich regulatory elements.

### Four types of regulation of chromatin accessibility by TFs

To investigate how Pou5f3, SoxB1 and Nanog create the regions of open chromatin, we restricted our analysis to the ARs, where any combination of the three factors was required for chromatin accessibility ("down" in MZ*triple*, Fig. 2a). To study only direct TF effects, we further selected 20131 TdARs (TF-bound Accessible Regions, down-regulated in the MZ*triple* mutant), overlapping with ChIP-seq peaks for Pou5f3, SoxB1[12], or Nanog[23].

We then classified TdARs by non-redundant requirements for Pou5f3 and Nanog for chromatin accessibility into four groups (Fig. 3a). Accessibility of the 1.PN group TdARs was reduced in MZ*spg* and MZ*nanog*, so we concluded that Pou5f3 and Nanog were both required in these regions. Pou5f3 was required in the 2.P group, Nanog in the 3.N group, and none of the single factors was required in the 4.– group (i.e. several factors were required redundantly). Out of the four groups, 2.P and 3.N were the most enriched for the Pou5f3- and Nanog – motifs, and the most highly occupied by Pou5f3 and Nanog, respectively (Fig. 3b, c).

To address if chromatin becomes accessible in all groups at the same time, we performed ATAC-seq at major ZGA (3 hpf), and compared ATAC-seq signals at 3, 3.7, and 4.3 hpf, in the wild-type and MZ*triple*. While all groups depended on PSN already at ZGA, chromatin in 4.– group was more accessible in MZ*triple* in all stages than in the other groups (Fig. 3d, dashed lines); Four groups were also significantly different in GC content, which increased in order 1.PN < 2.P < 3.N < 4.–, (Fig. 3e).

We next wondered whether the factors required for chromatin accessibility in a given group of TdARs were also sufficient to open chromatin. We used rescue experiments of Miao et al., 2022[5] to answer this question. To address which factors can restore chromatin accessibility in the triple mutants MZ*nps*, the authors microinjected Pou5f3, Sox19b, and Nanog mRNAs, individually and in combinations, and compared ATAC-seq signals in the injected and non-injected MZ*nps* embryos. Using their data, we scored the chromatin accessibility rescue in each of the four groups. Nanog alone rescued more than 90% of TdARs in the 3.N group. Pou5f3 alone rescued only 26% of TdARs in the 2.P group; most of the rest could be rescued by the Pou5f3/Sox19b or Pou5f3/Nanog combinations (Fig. 3f, g). Further, in 59% of TdARs in the 2.P group chromatin accessibility was also downregulated in the double Sox19b/Nanog mutant MZ*sn* (Fig. S3a, b). Thus, although Pou5f3 was present in MZ*sn*, it was apparently not sufficient to open chromatin in most of its target regions. We concluded that Nanog alone was required and sufficient for establishing chromatin accessibility in the majority of its target regions. In contrast, Pou5f3 needed the assistance of either Nanog or SoxB1 to establish accessible chromatin.

Figure 3h summarizes four types of establishment of chromatin accessibility by pioneer-like factors. Pou5f3 and Nanog are both required on the 1.PN group TdARs. Pou5f3 and redundant contribution of Nanog or SoxB1 are required on most of the 2.P group TdARs. Nanog is required on most of the 3.N group TdARs. Nanog, SoxB1 or to less extent Pou5f3 are redundantly required on 4.– group TdARs (see Fig. S3 d for detailed analysis of the group 4.–).

### Pou5f3 and Nanog act as activator and blocker

Pou5f3, SoxB1, and Nanog together promote histone acetylation on their binding sites, which is critical for the enhancer activation[5]. However, the individual roles of the TFs remain unclear. We used H3K27ac ChIP-seq data in the single mutants[17] (and this work) and

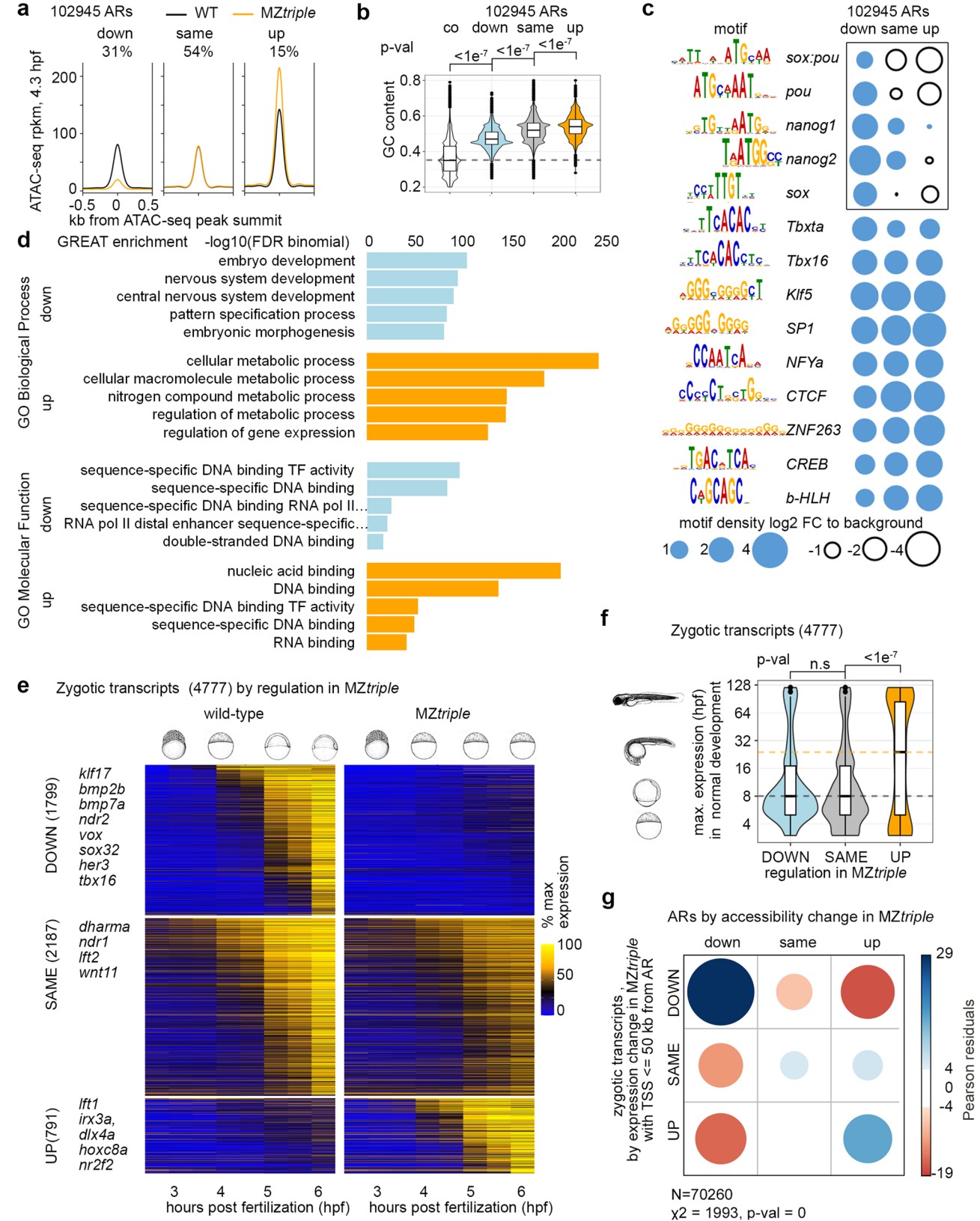

MZ*nps* triple mutant[5] to analyze how H3K27 acetylation relates to the pioneer-like activity of the factors.

The PSN combined activity was required for H3K27 acetylation in all groups, in agreement with published data[5] (Fig. 4a). Analyzing the single mutants, we found that the same TF which was strictly required for chromatin accessibility was required for H3K27 acetylation: both

Pou5f3 and Nanog in the 1.PN group, Pou5f3 in the 2.P group, Nanog in the 3.N group, and none of the single factors in the 4.– group (Fig. 4b). Overall GC content was lower in Pou5f3 - activated enhancers than in Nanog-activated enhancers (Fig. S4a, b). Unexpectedly, single mutant analysis revealed reciprocal antagonistic relationships between the TFs: Nanog reduced H3K27ac in the 2.P group, and Pou5f3 reduced

**Fig. 2 | Changes in chromatin accessibility parallel the changes in gene expression in Pou5f3, Sox19b and Nanog triple mutant. a** Three groups of accessible regions (ARs) were selected as follows: in "down" and "up" regions, ATAC-signal was reduced or increased, respectively, in six MZ*triple* biological replicates compared to seven wild-type biological replicates with FDR < 5%. "same" regions – all the remaining ARs which were considered unchanged. **b** ARs of "up" group had the highest GC content. *P*-values for two-sided Tukey-Kramer test; p-value for 1-way ANOVA was <2e$^{-16}$. $n$ = 102945 accessible regions derived from 6 independent ATAC-seq experiments. n "down"=31910, n"same" = 55439, n"up" = 15596, n "co" = 102944. To obtain control genomic regions (co, dotted line), genomic coordinates of all ARs were shifted 1 kb downstream **c** Pou5f3, Sox and Nanog-binding motifs (black rectangle) are enriched in "down" ARs. **d** GREAT analysis. **e** Three groups of zygotic transcripts by expression change in MZ*triple*. The heatmap shows normalized expression at eight time points, from pre-ZGA (2.5 hpf) till 6 hpf; example developmental genes at the left. **f** Group of zygotic genes is prematurely expressed in MZ*triple*. Y-axis: time of maximal expression in the normal development (3 hpf − 120 hpf from Ref. [10]), schematic embryo drawings illustrate the stages. Median expression time of transcripts upregulated in MZ*triple*

was 24 hpf (yellow dotted line), versus 8 hpf for down- or unchanged transcripts (gray dotted line). *p*-values in two-sided Tukey-Kramer test. "n.s" – non-significant $p = 0.1202769$ for the groups "DOWN" and "SAME". *p*-value in 1-way ANOVA was <2e$^{-16}$. $n$ "DOWN" = 1799, n"SAME" = 2187, n"up" = 791, the groups were derived from 3 wild-type and 3 MZ*triple* independent RNA-seq time curve experiments. **g** Down- or upregulation of chromatin accessibility in MZ*triple* correlates with respective transcriptional changes of linked genes. Two-sided $\chi^2$ test; positive correlations are shown in blue and negative in red. **b, f** The centers of the box plots correspond to the median values, the lower and upper bounds of the box correspond to the 25th and 75th percentiles, the upper whisker extends from the upper bound to the largest value no further than 1.5 * IQR (inter-quartile range), the lower whisker extends from the lower bound to the smallest value at most 1.5 * IQR. Outlying points beyond the end of the whiskers are plotted individually. Source data are provided as a Source Data file 1 (**a−d**), Source Data file 2 (**e, f**) and Source Data file 3 (**g**). Zebrafish embryo drawings were used with permission of John Wiley & Sons - Books, from "Stages of Embryonic Development of the Zebrafish", Kimmel et al., Developmental Dynamics 203:253-310 (1995); permission conveyed through Copyright Clearance Center, Inc.

H3K27ac in the 3.N group (Fig. 4b). Scoring the enhancers on which H3K27ac was induced (+), unchanged (0), or reduced (−) by each TF uncovered p + n- and p-n+ types of "antagonistic enhancers", on which Pou5f3 and Nanog regulated H3K27ac in the opposite directions (Fig. S4c-f, H3K27ac summary profiles in Fig. 4c, heatmaps in Fig. S5a). Sox19b regulated H3K27ac in the same direction as Pou5f3 on most of the antagonistic enhancers (Fig. S5b).

To understand how per se transcriptional activators and pioneer-like factors could negatively regulate enhancer activity, we took a closer look on chromatin accessibility changes on the antagonistic enhancers. We found that the factor which reduced H3K27ac also reduced chromatin accessibility (Fig. 4d, heatmaps in Fig. S5c, statistics in Fig. S5e). We confirmed this result using MNase-seq nucleosome positioning data[22] (Fig. S5d, e).

Together, we have shown by two independent methods that only one of the TFs, either Pou5f3 or Nanog, acts as pioneer-like factor on the antagonistic enhancers. This observation suggested that Pou5f3 and Nanog alternate their binding mode, acting as pioneer-like factors on some but not all genomic sites. The non-pioneer-like TF binding reduces chromatin accessibility, perhaps by competing for a common site with the pioneer-like TF. In the 2.P p + n- enhancer example within *her3* regulatory region, all three factors bind *sox:pou* motif; Pou5f3 and Sox19b binding induces chromatin accessibility and H3K27ac, Nanog binding reduces both (Fig. 4e, blue shading). In the 3.P p-n+ enhancer example within *morc3b* regulatory region, Nanog and Pou5f3 bind on two nearby *nanog* motifs, Nanog binding induces chromatin accessibility and H3K27ac, Pou5f3 binding reduces both (Fig. 4f, yellow shading).

Our next question was whether the positive and negative effects of Pou5f3 and Nanog on chromatin accessibility correlate with sequence features of the bound sites other than GC content. The open regions bound by both factors usually contained only one motif (Fig. S5f). As shown in Fig. S5g, Pou5f3 induced chromatin accessibility on its own motifs and reduced it on Nanog motifs, and vice versa. We hypothesized that Pou5f3 and Nanog can cross-recognize their motifs and compete for binding, while exact match to the motif and GC content around the motif matter for the pioneer-like activity.

Summarizing our findings, we suggest the general mechanism of antagonistic interactions between Pou5f3 and Nanog. Two TFs, A (activator) and B (blocker) compete for binding on the common motif within an enhancer (Fig. 4g). The activator acts as a pioneer-like factor on this motif: it displaces nucleosomes and promotes histone acetylation. Non-pioneer-like binding of the blocker protects the motif from the activator, thereby reducing nucleosome displacement and histone acetylation.

## Pou5f3 and Nanog bind DNA in a mutually exclusive way

Thse activator-blocker model assumes that 1) Pou5f3 and Nanog recognize shared motifs, and that 2) both TFs cannot bind the same motif at the same time. To test these assumptions, we selected 15 oligos from different enhancer types, on which Pou5f3 and Nanog either had opposite effects in chromatin accessibility (2.P + N- and 3.N + P- as in the examples in Fig. 4e, f), or both increased accessibility (1.PN group enhancers, renamed as 1.P + N+ for clarity). Fourteen 20-23 bp long oligos contained single motif hit, 28 bp long oligo 4 hit two motifs. We performed gel retardation assays with the labelled oligos in the presence of FLAG-tagged Pou5f3, HA-tagged Nanog, or both proteins for each oligo. In standard conditions, we detected moderate to strong binding of at least one protein in 10 cases of 15 (Fig. S6a). Five oligos sharing consensus sequence ATG[CT][TA]AAT strongly bound Pou5f3, four of them also bound Nanog with weaker affinity (Fig. 5a, b, d, Fig. S6b, c). Four oligos sharing consensus sequence T[GA]ATGG strongly bound Nanog, at least one of them (oligo 6) also weakly bound Pou5f3 (Fig. 5c, d, Fig. S6d for long exposure times). Both TFs interchangeably bound two motifs in the oligo 4 (Fig. S6e). In the reaction mixes containing both proteins, no increase of binding and no additional DNA-protein complexes were observed, compared to the mixes with one binding protein (compare the wells 1 and 2, 4 and 8 in Fig. 5a, b and Fig. S6 b, c; compare the wells 6 and 8, 11 and 12 in Fig. 5c and Fig. S6d). Moreover, increasing the concentration of Pou5f3 protein inhibited the formation of Nanog-DNA complexes and vice versa (Fig. S7). Figure 5e shows the explanation for mutually exclusive binding: Pou5f3- and Nanog motifs overlap in homeodomain-binding part, so only one of the proteins can contact DNA at the time. We also note that our in vitro binding experiments did not distinguish between 1.P + N+ synergistic and 2.P + N-antagonistic enhancers (compare Fig. S6c, d). We assumed that Nanog binding on Pou5f3 motifs may either promote (i.e. by priming the motif and facilitating subsequent Pou5f3 binding) or block Pou5f3 activity in vivo, depending on cell-specific cofactors and variations in GC content. In sum, in vitro binding experiments validated the idea that Pou5f3 and Nanog occupy shared motifs in a mutually exclusive way.

## Antagonistic enhancers balance zygotic gene expression

We wondered next, whether the antagonistic enhancers are relevant for zygotic gene expression. To answer this question, we chose the strategy to on the one hand sort zygotic transcripts into groups according to their regulation, without involving genomic information. On the other hand, we linked enhancers to transcripts and determined correlations between the enhancer and transcript groups.

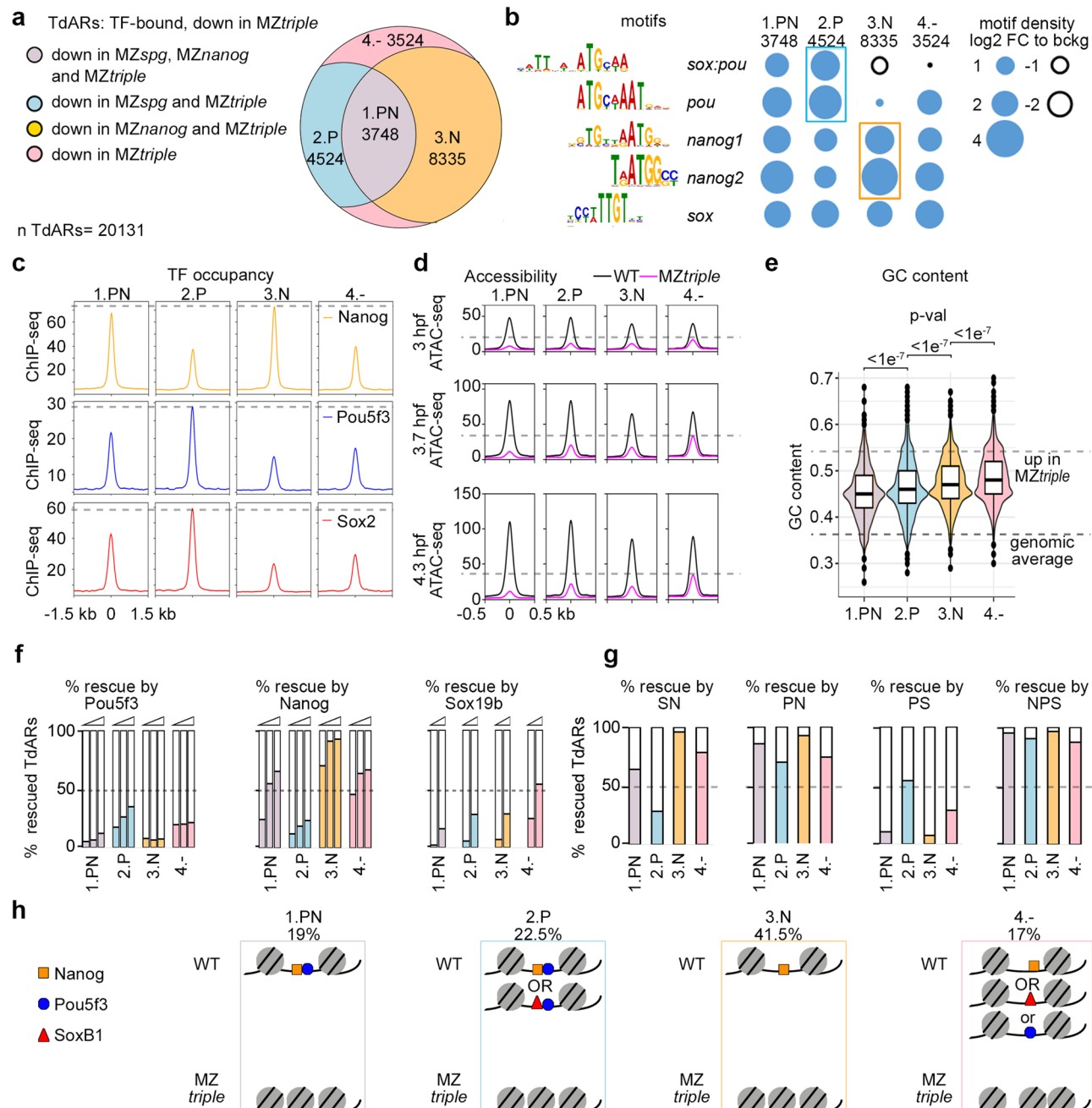

**Fig. 3 | Four types of TdARs by pioneer-like activity of Pou5f3, SoxB1, and Nanog. a** Four groups of TdARs by non-redundant requirements for Pou5f3 and/or Nanog for chromatin accessibility. **b** Frequencies of PSN motifs in the 4 groups. Note that 2.P and 3.N groups are the most enriched in the Pou5f3- and Nanog-specific motifs, respectively (colored boxes). **c** 2.P and 3.N groups are the mostly occupied by Pou5f3/SoxB1 and Nanog, respectively (gray dashed lines). Summary ChIP-seq profiles for indicated TFs (rpkm). **d** PSN establish accessible chromatin starting from major ZGA (3 hpf). ATAC-seq summary profiles in WT and MZ*triple* at 3, 3.7, and 4.3 hpf (rpkm). Note that chromatin in group 4.- regions in MZ*triple* is more accessible than in the other groups (gray dashed line). **e** GC content in four groups is significantly different: note that Pou5f3 is required in two groups with the lowest GC. *p* values for two-sided Tukey-Kramer test; *p*-value for 1-way ANOVA is <2e$^{-16}$. Groups were derived from at two to four independent ATAC-seq experiments in each of the four genotypes. n "1.PN" = 3748, n "2.P" = 4524, n "3.N" = 8335, n "4.-" = 3524. The lower dashed line shows the median genomic control GC content, upper line – median GC content of ARs upregulated in MZ*triple*. The centers of the box plots correspond to the median values, the lower and upper bounds of the box correspond to the 25th and 75th percentiles, the upper whisker extends from the upper bound to the largest value no further than 1.5 * IQR (inter-quartile range), the lower whisker extends from the lower bound to the smallest value at most 1.5 * IQR. Outlying points beyond the end of the whiskers are plotted individually. **f** Percentages of TdARs, for four groups, on which chromatin accessibility could be rescued by microinjection of different concentrations of single TFs into 1-cell stage MZ*nps* embryos[5]. RNA concentrations: low, normal, high for Pou5f3 and Nanog normal and high for Sox19b[5]. **g** Percentages of TdARs, for four groups, which could be rescued by microinjection of double or triple combinations of P,S and N TFs into 1-cell stage MZ*nps* embryos[5]. TFs in normal concentration. **h** Schematic drawing of four types of regulation of chromatin accessibility, percentage of each group from all TdARs is shown. "OR" – logical operator. Nanog and Sox19b are redundantly required for more regions within group 4.-, than Pou5f3. Source data are provided as a Source Data file 1 (**a**–**g**).

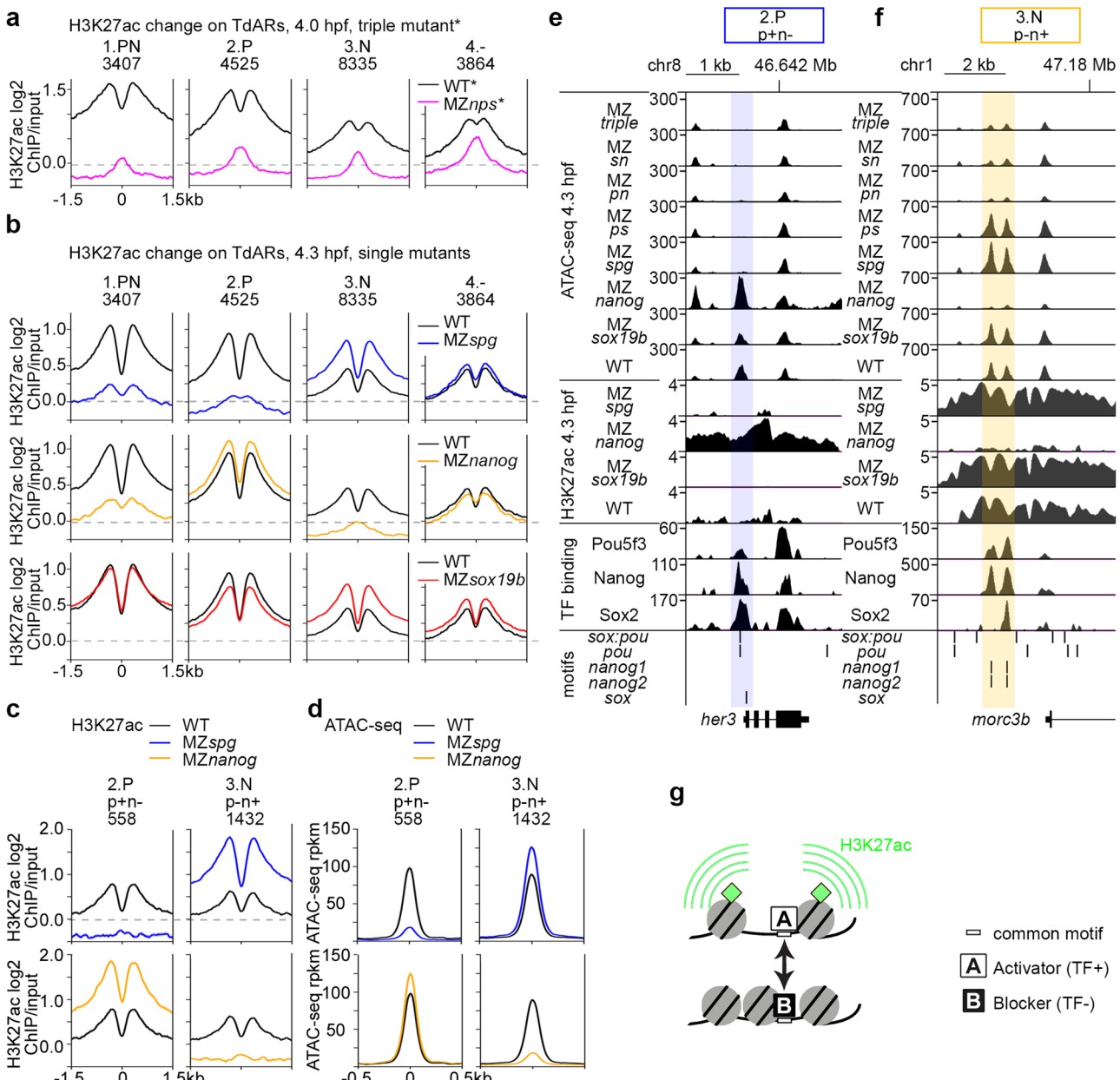

**Fig. 4 | Activator-blocker model: Pou5f3 and Nanog oppose each other effects on H3K27 acetylation and chromatin accessibility on a fraction of enhancers.** **a** Summary profiles of H3K27ac mark in the WT and MZ*nps* *-data from[5]. **b** Summary profiles of H3K27ac histone mark in the wild-type and single mutants, in four types of TdARs: 1.PN, 2.P, 3.N, 4.-. Note the opposite effects of Pou5f3 and Nanog on H3K27 acetylation on the 2.P and the 3.N groups. **c** Pou5f3 and Nanog have the opposite effects on H3K27ac on 2.P p + n- and 3.N p-n+ antagonistic enhancers. **d** Pou5f3 and Nanog have the opposite effects on chromatin accessibility on 2.P

p + n- and 3.N p-n+ antagonistic enhancers. **e, f** Genomic browser views show (from bottom to top) motif occurrence, TF binding, H3K27ac and ATAC-seq in the indicated genotypes. **e** All three TFs bind to *sox:pou* motif on *her3* 2.P p + n- antagonistic enhancer (blue shading). **f** Pou5f3 and Nanog, or all three factors bind *nanog* motifs on *morc3b* 3.N p-n+ antagonistic enhancers (yellow shading). **g** Schematic illustration of activator-blocker model (explanations in the text). Source data are provided as a Source Data file 1 (**a**–**f**).

To sort the transcriptome, we used a mathematical modeling approach in two subsequent steps. In the first step, we developed a "core model" to describe the dynamic behavior of the known core components of the system: Pou5f3, Nanog, and SoxB1 group genes (including Sox19b, Sox19a, Sox3, and Sox2). In the second step, dynamics of these core components were then used to inform ODE-based mini models to formalize all possible regulatory inputs from Pou5f3, Nanog or SoxB1 on the individual target genes.

The core model was based on ordinary differential equations (ODEs), where initial assumptions on the regulatory interactions between the core components were taken from the literature

(Methods, Fig. S8a for model cartoon and Source Data file 5 for model equations). To infer the model's parameters, we used the measured time-resolved transcriptional profiles for *pou5f3*, *nanog*, *sox19b*, *sox19a*, *sox3* and *sox2*. After model calibration, the model trajectories resulting from the best fit parameters were in good agreement with the experimental data (Fig. S8b, c, Fig. S9a, and Source Data file 6).

For the second step of ODE-based mini models, we assumed that at least one factor of Pou5f3, Nanog or SoxB1 should activate the target, while the other two can activate (+), repress (−) or do nothing (0). This gives 19 possible regulatory combinations. Each of the 19 mini models was separately fitted to experimental time-resolved RNA-seq

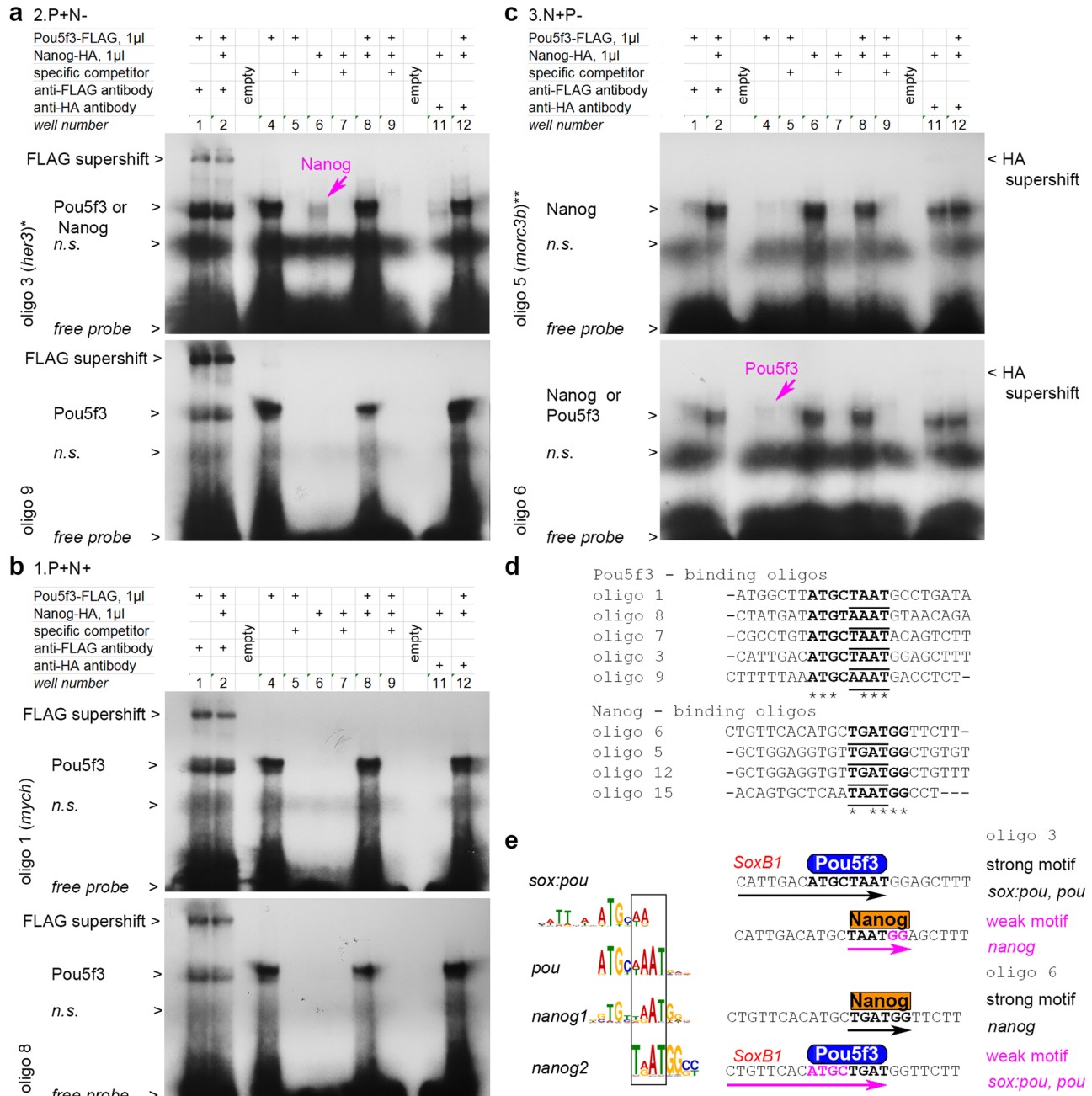

**Fig. 5 | Pou5f3 and Nanog bind in a mutually exclusive manner to overlapping motifs.** a–c Gel-retardation assays with the indicated oligos.*, ** - genomic locations oligo 3 and oligo 5 are shown in Fig. 4e, f. **a** Pou5f3-binding oligos from 2.P + N− antagonistic enhancers. Weaker Nanog binding is also detectable for oligo 3 (magenta arrow); see Fig. S6b with longer exposure time for Nanog-HA supershift with oligo 3 and Nanog binding to oligo 9. **b** Pou5f3-binding oligos from 1.P + N+ synergistic enhancers (oligo 1 is from *mych*[55] enhancer, Nanog binding was not detectable). See Fig. S6c with longer exposure time for Nanog binding to oligo 8 and oligo 7. **c** Nanog- binding oligos from 3.N + P- antagonistic enhancers. Weaker

Pou5f3 binding is also detectable for oligo 6 (magenta arrow). See Fig. S6d with longer exposure times for Nanog-HA supershifts. **d** Aligned Pou- and Nanog- strong consensus binding sequences from our assays (in bold); homeodomain-binding part is underlined. **e** Left: Pou5f3 and Nanog binding motifs share common part, recognized by homeodomains (black box). Right: in oligo 3 and oligo 6, *pou* and *nanog* motifs overlap in homeodomain-recognition part, so that only Pou5f3 or only Nanog can contact DNA at the same time. SoxB1 binding part of *sox:pou* motif[20] is indicated in red. *n.s* − non-specific band. Source data are provided as a Source Data file 4.

data for each of the 1799 zygotic transcripts that had been found to be directly or indirectly activated by Pou5f3, Sox19b or Nanog ("DOWN" group in Fig. 2e, Source Data file 7). The dynamics of Pou5f3, Nanog, and SoxB1 group genes (SOX) obtained in step one of the analysis were used as input to the mini models, while the remaining parameters were again calibrated based on the RNA-seq data. Fits of 18 non-empty mini-models are shown for exemplary targets in Fig. S9b. To simplify the analysis with respect to our goal of connecting chromatin state and

transcription for Pou5f3 and Nanog, we merged the 19 mini-models down to six mini-model groups capturing all possible variants of target regulations by Pou5f3 and Nanog and the regulation by SOXB1 alone (Fig. 6a).

For each transcript, we selected the best candidate mini-model group according to Bayesian Information Criterion (BIC, see Source Data file 8 and Source Data file 6). Fits of selected mini-model groups are shown for exemplary targets in Fig. 6b. 1799 transcripts were then

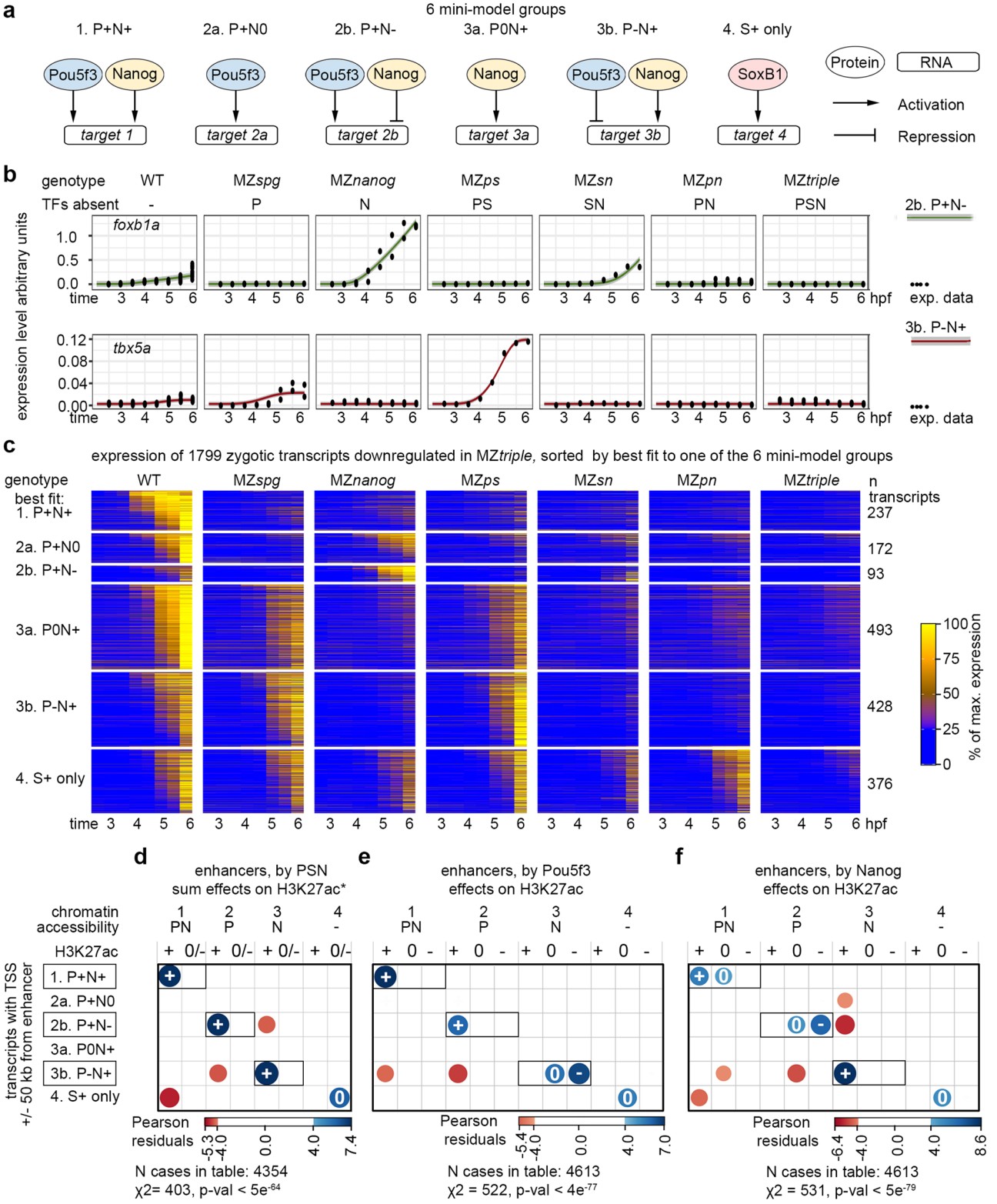

**Fig. 6 | Zygotic gene expression is balanced by synergy and competition of Pou5f3 and Nanog on common enhancers. a** Schemes of six alternative mini-model groups for direct target regulation by Pou5f3, Nanog and SoxB1. **b** *foxb1a* and *tbx5a* transcripts dynamics fitted best to the antagonistic groups 2b. P + N- and 3b. P-N+ respectively. **c** Heatmap of zygotic genes, downregulated in MZ*triple*, sorted by best fit to one of the six model groups. **d–f** Two-sided χ² tests, positive correlations between the transcriptomics and genomics groups are shown in blue and negative in red. Exact *p*-values: *p* = 4.292e-64 (**d**), *p* = 3.128e-77 (**e**), *p* = 4.314e-79 (**f**). Vertical axis: transcriptomics groups. Direct enhancers linked to the promoters of zygotic genes were sorted by best fit transcriptional model. Synergistic and antagonistic model groups are boxed. Horizontal axis: direct enhancers were sorted by chromatin accessibility, and by H3K27ac regulation by sum of PSN activities (**d**), by Pou5f3 (**e**), or by Nanog (**f**). *- data from[5]. Source data are provided as a Source Data file 2 (a-c) and Source Data file 3 (**d–f**).

classified into six groups according to the best fitting direct regulatory scenario (Fig. 6c).

To inquire if synergistic and antagonistic chromatin regulation by Pou5f3 and Nanog on common enhancers is directly linked to synergistic and antagonistic regulation of transcription, we put together the results of genomic and transcriptomic analyses. As shown in Fig. 6d–f, the predicted direction of transcriptional regulation of targets by Pou5f3 and Nanog (activation or repression) strongly correlated with Pou5f3 and Nanog-dependent H3K27ac changes on their enhancers (activation or blocking), respectively (see also Fig. S10a, b). The enhancers linked to Pou5f3 or Nanog – activated transcript groups were enriched in respective motifs and had significantly different GC content (Fig. S10c, d). Thus, cross-validation of independent genomic and transcriptomic data sets demonstrated that the transcription at ZGA is directly regulated by synergistic and antagonistic enhancers.

### Pou5f3-Nanog antagonism blocks premature transcription

We have noticed that all mutants except MZ*sox19b* prematurely transcribe poorly overlapping sets of genes, enriched for DNA-binding and transcription regulatory functions (Fig. S2). We have also demonstrated that Pou5f3 and Nanog compete on antagonistic enhancers as activators and blockers. Putting these two findings together, we wondered again whether Pou5f3 could directly block premature transcriptional activation by Nanog and vice versa. To answer this question, we examined putative regulatory regions of the transcripts, upregulated in the Pou5f3 and Nanog mutants, for the enrichment of antagonistic enhancers of the opposite types.

In Pou5f3 mutant MZ*spg*, 17% of transcripts were upregulated compared to the wild-type (Fig. 7a) and contained prematurely expressed genes (Fig. 7b). Strikingly, the putative regulatory regions of these genes were highly enriched for antagonistic 3.N p-n+ enhancers (Fig. 7c, boxed). 3.N p-n+ enhancers were also enriched in the putative regulatory regions of genes upregulated in the double Pou5f3/Sox19b mutant MZ*ps* (Fig. S10 e–g). Thus, Pou5f3 blocked inappropriate transcriptional activation of late developmental genes by Nanog.

To test if the reverse was also true, we examined the transcripts upregulated in MZ*nanog*. 10% of transcripts were upregulated compared to the wild-type in MZ*nanog* and contained prematurely expressed genes (Fig. 7d, e). The putative regulatory regions of these genes were enriched for antagonistic 2.P p+n- enhancers (Fig. 7f, boxed). Thus, Nanog blocked inappropriate transcriptional activation of late developmental genes by Pou5f3.

In sum, cross-validation of independent genomic and transcriptomic data sets demonstrated that the transcription of PSN target genes at ZGA is directly regulated by synergistic and antagonistic enhancers. On the synergistic enhancers, Pou5f3 and Nanog are both required to establish chromatin accessibility and activate early zygotic genes (Fig. 8a). On the antagonistic enhancers, "blocker" TF(−) attenuates activation of the early zygotic genes by "activator" (TF + ), or prevents the activation of lineage-specific regulators by "activator" TF (Fig. 8b, c).

## Discussion

Our study provides the mechanistic links between the pioneer-like activity of genome activators Pou5f3, Sox19b (together with zygotic SoxB1 factors) and Nanog, and their potential to activate transcription at ZGA and during the first half of gastrulation. We demonstrate that direct binding of PSN to genome is not only responsible for transcriptional activation at ZGA, but also for adjusting the levels of early zygotic transcripts and for the timing of gene expression in development.

Synergistic interactions were long ago suggested for zebrafish genome activators and their mammalian homologs[12,15]. Antagonistic interactions of Pou5f3 and Nanog on chromatin were not described before. A reasonable question is which features define activator and blocker functions in each case. We found a correlation with two

sequence features. One feature is GC content around TF binding motifs: GC content was generally lower on the regions activated by Pou5f3 than on the regions activated by Nanog (Fig. 3e, Fig. S4). Another feature is the different frequency of sequence-specific motifs: we found that a given TF acts more frequently as activator on its own motifs, and as a blocker as it binds to the motifs of the other factor (Fig. 5, Fig. S4g). In a sense, "activator" and "blocker" TFs can be compared with a precise key and a key blank to the locked door (which is an enhancer): both keys fit to the lock, but only a precise key can open the door (activate an enhancer).

Other than that, we could not find the mechanistic differences between TF binding on synergistic and antagonistic enhancers: in both cases Pou5f3 and Nanog likely bind in turn to the common motifs (Fig. 5e). In whole embryo assays, we could not distinguish whether the ATAC-seq signals come from the whole embryo or from the subpopulation of cells. Therefore, we assume that cell-specific differences, such as local concentrations of TFs, presence of transcriptional cofactors or crosstalk with signaling pathways affect the binding mode cell type-specifically. In line with that, it was shown that nucleosome displacement activity of known mammalian pioneer factors, including Oct4/Pou5f1, is conditional: all of them bind to different genomic locations in different cell types[24]. Further, it is also possible that antagonistic enhancers act as bifunctional cis-regulatory elements, i.e. as activators or as silencers, depending on the embryonic cell type[25]. Single cell multi-omics and reporter assays are needed to clarify this issue.

The dual role of Pou5f3 and Nanog in activating early zygotic genes and blocking premature transcription provides an immediate parallel to the biological roles of their mammalian homologues. Oct4/Pou5f1 and Nanog regulate the seemingly opposite processes of pluripotency maintenance and cell lineage decisions in mammalian embryos and ES cells[26–30]. Although zebrafish Pou5f3 is not equivalent to Oct4/Pou5f1 in terms of nucleosome-binding and reprogramming capacity[31], Pou5f3 and Oct4 recognize the same motifs and colocalize with Nanog in the respective systems. It is tempting to speculate that Oct4 and Nanog may reciprocally block some of each other activities similarly to their zebrafish homologues. Recent study supports this possibility: acute depletion of Oct4 in ES cells results in increased genomic binding of Nanog, suggesting that Oct4 outcompetes Nanog on chromatin and antagonizes its function[32].

Our results suggest that the list of zebrafish genome activators is not complete: hundreds of transcripts, enriched in transcriptional regulatory functions, are prematurely expressed in MZ*triple* mutant and linked to regulatory elements with high GC content (Fig. 2). Identification of putative GC-binding factors regulating these genes and dissecting their interplay with Pou5f3, SoxB1 and Nanog will further clarify the mechanisms of gene regulation at ZGA. Finally, the concepts presented in this work will assist the studies of combinatorial genome activation by TFs in other organisms and deepen our understanding of two global and currently unresolved questions: conditional activity of pioneer and pioneer-like factors in different biological settings, and of how the gene expression is regulated in developmental time.

## Methods

All experiments were performed in accordance with German Animal Protection Law (TierSchG) and European Convention on the Protection of Vertebrate Animals Used for Experimental and Other Scientific Purposes (Strasburg, 1986). The generation of double and triple mutants for Pou5f3, Sox19b and Nanog was approved by the Ethics Committee for Animal Research of the Koltzov Institute of Developmental Biology RAS, protocol 26 from 14.02.2019.

### Experimental model and subject details
**Zebrafish maintenance and generation of the double and triple mutants.** Wild-type fish of AB/TL and mutant strains were raised,

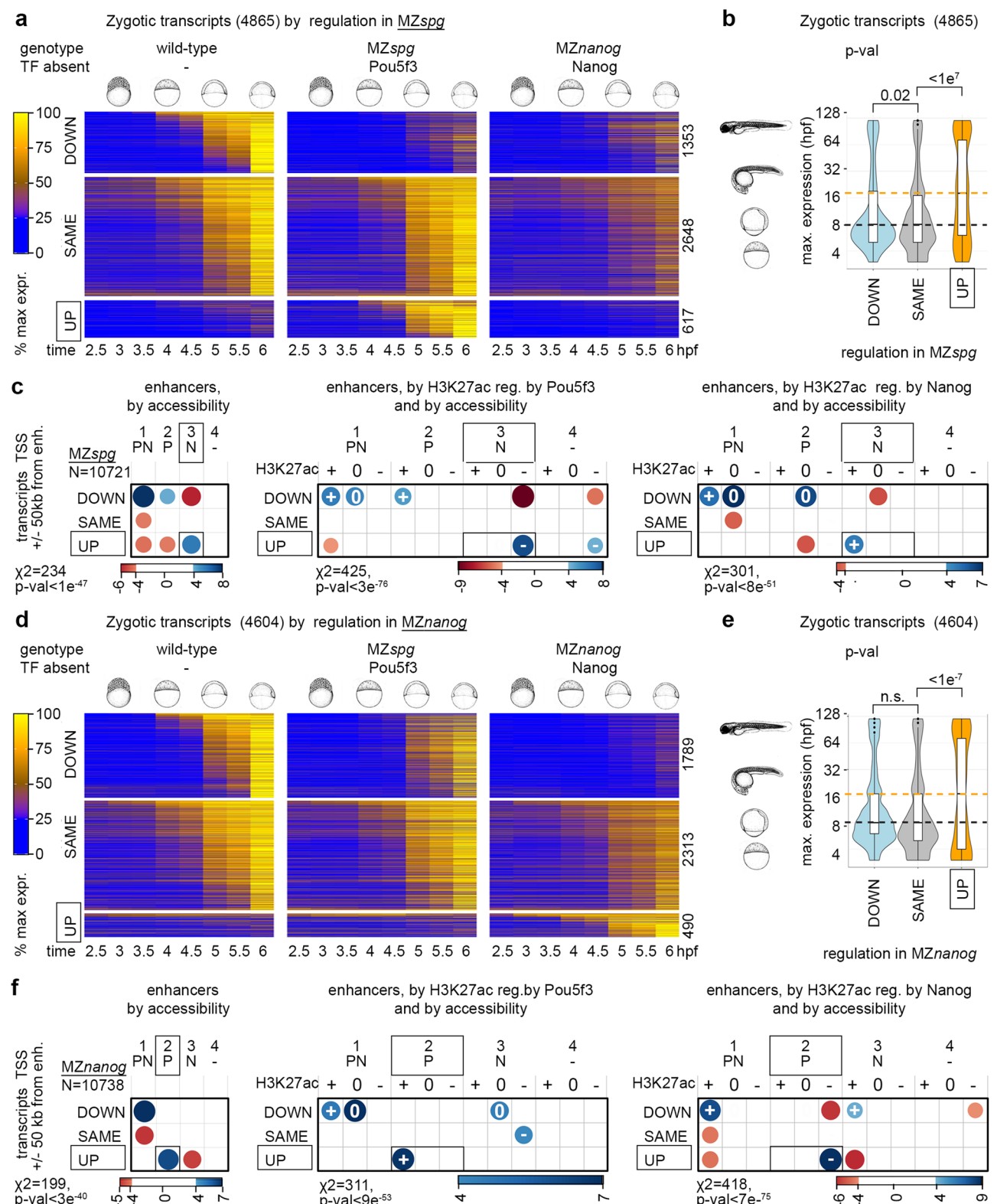

maintained and crossed under standard conditions as described by Westerfield[33]. Embryos were obtained by natural crossing (4 males and 4 females in 1,7l breeding tanks, Techniplast), from fish aged from 3 months to 2 years. Wild-type and mutant embryos from natural crosses were collected in parallel in 10-15 minutes intervals and raised in egg water at 28.5 °C until the desired stage. Staging was performed following the Kimmel staging series[34]. Stages of the mutant embryos were indirectly determined by observation of wild-type embryos born

at the same time and incubated under identical conditions. Sex of the embryos was not considered because sex determination in zebrafish occurs in later developmental stages and zebrafish do not have sex chromosomes. All experiments were performed in accordance with German Animal Protection Law (TierSchG) and European Convention on the Protection of Vertebrate Animals Used for Experimental and Other Scientific Purposes (Strasburg, 1986). The generation of double and triple mutants for Pou5f3, Sox19b and Nanog was approved by the

**Fig. 7 | Pou5f3 blocks premature transcriptional activation by Nanog and vice versa. a–c** Pou5f3 blocks transcription from Nanog-dependent enhancers. **d–f** Nanog blocks transcription from Pou5f3-dependent enhancers. Zygotic transcripts were sorted by expression change in MZ*spg* (**a**) or MZ*nanog* (**d**) compared to the wild-type. Heatmaps show normalized expression at eight time points, from pre-ZGA (2.5 hpf) till 6 hpf; numbers of transcripts in each group are shown at the right. **b, e** Premature expression of zygotic genes in MZ*spg* (**b**) or in MZ*nanog* (**e**). Median expression time in the normal development (3 hpf − 120 hpf from ref. [10]) was compared in "UP" "DOWN" and "SAME" groups. p-values in Tukey-Kramer test; p-value in 1-way ANOVA was <2e$^{-16}$. "n.s" – non-significant $p = 0.087583$ for the groups "DOWN" and "SAME" in (**e**). The centers of the box plots correspond to the median values, the lower and upper bounds of the box correspond to the 25th and 75th percentiles, the upper whisker extends from the upper bound to the largest value no further than 1.5 * IQR (inter-quartile range), the lower whisker extends from the lower bound to the smallest value at most 1.5 * IQR. Outlying points beyond the end of the whiskers are plotted individually. **b** n "DOWN" = 1353,

n"SAME" = 2648, n"UP" = 617, the groups were derived from 5 wild-type and 3 MZ*spg* independent RNA-seq time curve experiments. **e** n "DOWN" = 1789, n"SAME" = 2313, n"up" = 490, the groups were derived from 5 wild-type and 3 MZ*nanog* independent RNA-seq time curve experiments. **c, f** Two-sided $\chi^2$ tests, positive correlations between the transcriptomics and genomics groups are shown in blue and negative in red, scales show Pearson residuals. Exact p-values from left to right: $p = 9.391e\text{-}48$, $p = 2.873e\text{-}76$, $p = 7.739e\text{-}51$ (**c**), $p = 2.574e\text{-}40$, $p = 8.064e\text{-}53$, $p = 6.162e\text{-}75$ (**f**). Vertical axis: transcriptomics groups. Direct enhancers were linked to the promoters of zygotic genes within +/−50 kb and sorted according to DOWN", "SAME" and "UP" transcriptional groups. Horizontal axis: genomic groups. Direct enhancers were sorted by changes in chromatin accessibility, and by H3K27ac regulation by Pou5f3 or Nanog, as indicated. Source data are provided as Source Data file 2 (**a, b, d, e**) and Source Data file 3 (**c, f**). Zebrafish embryo drawings were used with permission of John Wiley & Sons - Books, from "Stages of Embryonic Development of the Zebrafish", Kimmel et al., Developmental Dynamics 203:253-310 (1995); permission conveyed through Copyright Clearance Center, Inc.

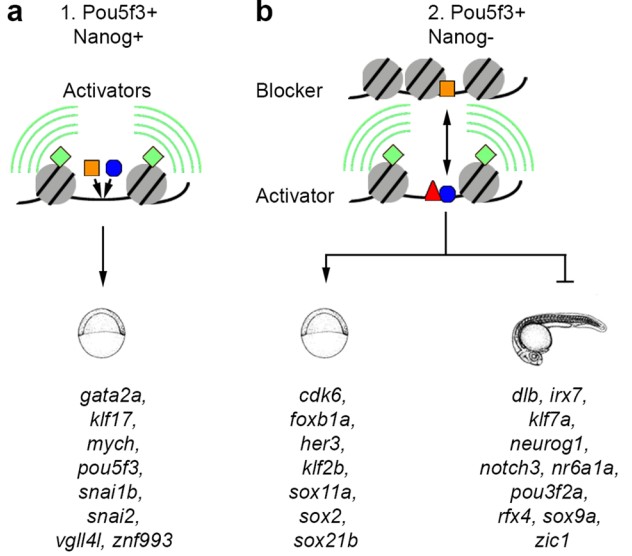
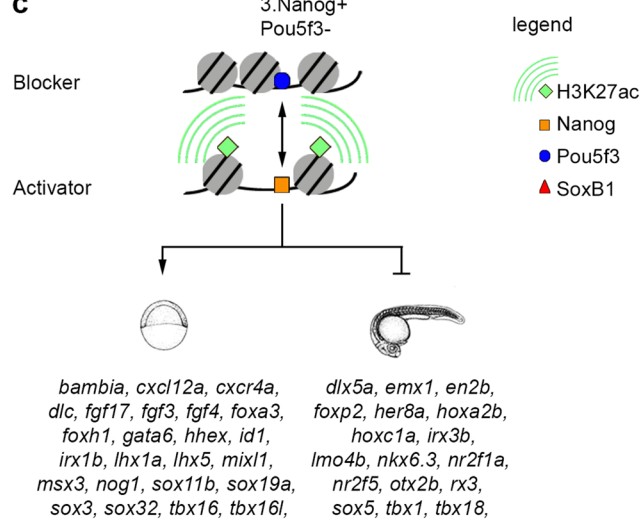

**Fig. 8 | Synergistic and antagonistic Pou5f3/Nanog direct enhancers: the mechanism of action and direct target genes. a** Synergistic enhancers Pou5f3+Nanog + : Pou5f3 and Nanog act as activators. **b** Antagonistic enhancers Pou5f3+Nanog-: Pou5f3 acts as an activator, Nanog as a blocker. **c** Antagonistic enhancers Pou5f3-Nanog + : Nanog acts as an activator, Pou5f3 as a blocker. Representative early targets (expressed in the wild-type during 2.5-6 hpf time course), and late targets (prematurely expressed in the mutants by blocker TF

during 2.5-6 hpf time course) are indicated by schematic early and late embryo drawings. Source data are provided as a Source Data file 3. Zebrafish embryo drawings were used with permission of John Wiley & Sons - Books, from "Stages of Embryonic Development of the Zebrafish", Kimmel et al., Developmental Dynamics 203:253-310 (1995); permission conveyed through Copyright Clearance Center, Inc.

Ethics Committee for Animal Research of the Koltzov Institute of Developmental Biology RAS, protocol 26 from 14.02.2019. Maternal-Zygotic (MZ) homozygous mutant embryos MZ*sn* were obtained in three subsequent crossings. First, MZ*nanog*$^{m1435}$ [19] homozygous males were crossed with MZ*sox19b*$^{m1434}$.[17] homozygous females. The double heterozygous fish were raised to sexual maturity and incrossed. The progeny developed into phenotypically normal and fertile adults. Genomic DNA from tail fin biopsies was isolated and used for genotyping. We first selected *sox19b* homozygous mutants, by PCR-with Sox19b-f1/Sox19b-r1 primers followed by restriction digest with BbsI[17]. To select the double homozygous fish, we used the genomic DNA from *sox19b* homozygous mutants for by PCR with Nanog1-f1/Nanog1-r1 primers followed by restriction digest with NdeI. MZ*sn* embryos for experiments were obtained from incrosses of double homozygous *sox19b*-/-; *nanog*-/- fish. The line was maintained by crossing *sox19b*-/-; *nanog* -/-; males with *sox19b*-/-; *nanog* +/- females.

MZ*pn* homozygous mutant embryos were obtained in three subsequent crossings. First, MZ*nanog* $^{m1435}$ homozygous males[19] were crossed with MZ*spg*$^{793}$ [18] homozygous females. The double

heterozygous fish were raised to sexual maturity and incrossed. To bypass the early requirement for Pou5f3 in the spg*793* homozygous mutants, one-cell stage embryos were microinjected with 50p-values100 pg synthetic Pou5f3 mRNA as previously described[18]. The fish were raised to sexual maturity, genomic DNA from tail fin biopsies was isolated and used for genotyping. We first selected *nanog* homozygous mutants, by PCR with Nanog1-f1/Nanog1-r1 primers followed by restriction digest with NdeI. To select the double homozygous fish, we used the genomic DNA from *nanog* homozygous mutants to PCR-amplify the region, flanking the *spg*$^{m793}$ allele. *Spg*$^{m793}$ allele carries an A- > G point mutation in the splice acceptor site of the first intron of Pou5f3 gene, which results in the frameshift starting at the beginning of the second exon, prior to the DNA-binding domain. *Spg*$^{m793}$ is considered to be null allele. We used the following PCR primers: spg-f1 5'-' GTCGTCTGACTGAACATTTTGC −3' and spg-r1 5'-' GCAGTGATTCTGAGGAAGAGGT −3'. Sanger sequencing of the PCR products was performed using commercial service (Sigma). The sequencing traces were examined and the fish carrying A to G mutation were selected. MZ*pn* embryos for experiments were

obtained from incrosses of double homozygous spg−/−; *nanog*-/- fish. The line was maintained by crossing spg−/−; *nanog* −/− males with spg−/−; *nanog* +/− females and microinjecting Pou5f3 mRNA in each generation.

To obtain the triple Maternal-Zygotic (MZ) homozygous mutant embryos MZ*triple*, MZ*sn* double homozygous males were crossed with MZ*ps* double homozygous females[17]. The *sox19b*-/-; spg + /-; *nanog* + /- progeny was incrossed, microinjected with 50-100 pg synthetic Pou5f3 mRNA and genotyped for spg and *nanog* as described above. MZ*triple* embryos for experiments were obtained from incrosses of triple homozygous fish. The line was maintained by crossing *sox19b*-/-; spg-/-; *nanog*-/- males with *sox19b*-/-; spg-/-; *nanog* + /- females and micro-injecting Pou5f3 mRNA in each generation.

**Genomic DNA isolation and PCR for genotyping.** Genomic DNA was isolated from individual tail fin biopsies of 3 months old fish. Tail fin biopsies or embryos were lysed in 50 μl lysis buffer (10 mM Tris pH 8, 50 mM KCl, 0.3% Tween20, 0.3% NP-40, 1 mM EDTA) and incubated at 98 °C for 10 min. After cooling down Proteinase K solution (20 mg/ml, A3830, AppliChem) was added and incubated overnight at 55 °C. The Proteinase K was destroyed by heating up to 98 °C for 10 min. The tail fin biopsies material was diluted 20x with sterile water. 2 μl of was used as a template for PCR. PCR was performed in 25-50 μl volume, using MyTaq polymerase (Bioline GmbH, Germany) according to the man-ufacturer instructions, with 30-35 amplification cycles. The primer sequences are listed in the Supplementary Table.

**Method details**

**ATAC-seq and library preparation.** Omni-ATAC-seq was performed as described in[17]. Briefly, embryos were enzymatically dechorionated with Pronase E (30 mg/ml). 30 embryos were deyolked manually with an eyebrow needle in Danieaus medium, centrifuged (500 x g, 5 min, 4 °C), washed with 100 μl ice-cold PBS, and then centrifuged again. The cells were resuspended by pipetting in 50 μl ice-cold lysis buffer (ATAC-RSB (10 mM TrisHCl pH 7.4, 10 mM NaCl, 3 mM MgCl2), with 0.1% NP40, 0.1% Tween20, and 0.01% digitonin) and incubated on ice for 5 minutes. Then, the lysate was diluted with ice-cold dilution buffer (1 ml, ATAC-RSB containing 0.1% Tween-20). Nuclei were pelleted (500 x *g*, 10 min, 4 °C), the supernatant carefully removed, and the nuclei resuspended in 16.5 μl PBST. Tagmentation reaction was assembled according to the manufacturer instructions for Illumina small Tn5 buffer and enzyme kit in 50 μl volume and incubated in a thermomixer for 30 min (37 °C, 800 rpm). The sample was purified using Qiagen PCR MinElute Purification kit. The purified transposed mix was pre-amplified for 5 cycles (72 °C for 5 min, 98 °C for 30 sec, 5 cycles of (98 °C for 10 sec, 63 °C for 30 sec, 72 °C for 1 min) using NEBNext 2x Master Mix (NEB) and Nextera adapters and put on ice. The number of additional cycles was determined using 5 μl of the pre-amplified mixture as in ref. [35]. The total number of PCR cycles was 10-12. The amplified libraries were purified using SPRI beads (Beckmann) according to the manufacturer's instructions. ATAC-seq was performed at three developmental stages, 3 hpf (1k), 3.7 hpf (oblong), and 4.3 hpf (dome). At 3 hpf, three biological replicates for WT and MZ*triple* were used. At 3.7 hpf, two biological replicates for MZ*nanog* and MZ*pn* and three biological replicates for MZ*triple* were used. At 4.3 hpf, three biological replicates for MZ*sn* and MZ*triple* and two for MZ*pn* and MZ*nanog* were used. Paired-end 150 bp sequencing of resulting 23 ATAC-seq libraries was performed on NovaSeq6000 (Illumina) by Novogene company, with the sequencing depth 80 mil-lion reads per library. The raw and processed ATAC-seq data are deposited in GEO (NCBI), accession number GSE215956.

**ATAC-seq data processing.** ATAC-seq data processing was per-formed as in[17] on the european Galaxy server[36]. Adapters were removed, the reads were cropped to 30 bp from start, the reads with average quality less that 30 and length less than 25 bp were removed using Trimmomatics. The trimmed reads were aligned to the danrer11/GRCz11 genome assembly without alternative contigs using Bowtie2[37], with the parameters: '−dovetail, --very-sensitive, --no-unal'. The aligned reads were filtered using BamTools Filter[38] with the parameters -isProperPair true, -mapQuality >=30, -isDuplicate false, -reference! chrM. The aligned reads were filtered further using the alignmentsieve tool from the deepTools2 suite[39], with the parameters −ATACshift, --minMappingQuality 1, --maxFragmentLength 110. Thus, only frag-ments with a size of 110 bp or less were kept. The filtering steps were performed for the individual replicates as well as and for the merged biological replicates BAM files. Bigwig files for each stage and genotype were obtained from merged BAM files using Bamcoverage and used for data visualization in deepTools2. Bigwig and peak files for each stage and genotype are included in GEO GSE215956 submission as processed files.

**Definition of regulated ARs and TdARs.** The ATAC-seq data from this work at 3.7 hpf and 4.3 hpf were processed together with our pre-viously published ATAC-seq data set: three biological replicates for the wild-type and two biological replicates for MZ*spg*, MZ*sox19b* and MZ*ps* at 3.7 hpf, three biological replicates for the wild-type, MZ*spg* and MZ*sox19b* and four for MZ*ps* at 4.3 hpf[17], GEO accession number GSE188364. For seven wild-type biological replicates, filtered reads were converted from BAM to BED using Bedtools[40] and were used as inputs for peak calling. Peaks were called from individual replicates and merged replicates using MACS2 callpeak[41] with the additional parameters '--format BED, --nomodel, --extsize 200, --shift −100' and a 5% FDR cutoff. Regions mapping to unassembled contigs were exclu-ded. Only peaks that were overlapping in merged and each of the single replicates were kept. ATAC-seq peaks overlapping between 3.7 and 4.3 hpf stages in the wild-type were centered on ATAC-seq peak summits (4.3 hpf) and cut to 110 bp length. This set of 102 965 regions accessible in the wild-type (ARs) was used in all subsequent analyses. The coverage of ATAC-seq reads on ARs was scored using Multi-CovBed (Bedtools) in all ATAC-seq experiments (3.7 and 4.3 hpf). Deseq2[42] was used to normalize the reads and compare chromatin accessibility in each mutant to the wild-type. For each mutant geno-type, we made four Deseq2 cross-normalizations: across all replicates, across the replicates of 3.7 hpf or 4.3 hpf separately, and across three samples (i.e. wild-type, MZ*nanog* and MZ*triple*). The region was scored as downregulated in the mutant, if the fold change to the wild-type was negative with FDR 5% in four Deseq2 normalizations. The region was scored as upregulated in the mutant, if the fold change to the wild-type was positive with FDR 5% in four Deseq2 normalizations. The remaining ARs were considered unchanged. To define TdARs, (TF-bound Accessible Regions, downregulated in MZ*triple*), we selected ARs, downregulated in MZ*triple* and overlapping with any of Sox2, Pou5f3[12] or Nanog[23] ChIP-seq peaks for at least 1 bp. The genomic coordinates of TF-binding peaks remapped to danrer11/GRCz11 assembly were taken from[17]. Genomic coordinates of ARs and their properties are provided in the Source Data file 1.

**Scoring the motif enrichment.** Sequence-specific *nanog1* and *nanog2* motif matrices were taken from[22], the rest of the motif matrices from[17]. Genomic coordinates of all occurrences for each motif in the 3 kb regions around ATAC-seq peaks were obtained using FIMO[43] with p-value threshold $10^{-4}$ and converted to BED files. To calculate the background frequency of the motifs, we used the background control peak file, where the genomic coordinates of all ATAC-seq summits were shifted 1 kb downstream. The motif densities were calculated as the average number of motifs overlapping with 60 bp around the ATAC-seq or control peak summits. The log2 ratio of ATAC-seq to background motif densities was taken as a motif enrichment value.

**GC content and in-vitro nucleosome predictions.** GC content for 110 bp around AR summits was calculated with geecee program, in-vitro nucleosome prediction with the algorithm of Kaplan et al.[44] in the european Galaxy instance usegalaxy.org. The genomic coordinates of ARs and control peaks were extended to +/−5 kb to account for the edge effects, and in-vitro nucleosome prediction value was derived for every base pair. Nucleosome predictions were converted to bedgraph files for scoring. GC content and nucleosome prediction values per 110 bp around ARs are provided in the Source Data file 1.

**Scoring the rescue of chromatin accessibility.** Track hub with Omni-ATAC rescue processed data in UCSC bigwig format was downloaded from the home page of Antonio Giraldez lab (https://www.giraldezlab.org/data/miao_et_al_2022_molecular_cell/) and used for scoring. We calculated ATAC-seq signals in all conditions for 110 bp around ARs. For ARs, downregulated in MZ*nps* compared to the WT B1 replicate in[5], we estimated the rescue. We considered AR as rescued by TF(s) microinjection, if the log2 of ATAC-seq signals (injected MZ*nps*/control MZ*nps*) was exceeding log2 of ATAC-seq signals (control WT B1/injected MZ*nps*). The rescue status by all combinations of TFs for ARs downregulated in MZ*nps* is provided in the Source Data file 1.

**ChIP-seq for H3K27ac in MZ*nanog*.** The ChIP-seq was performed as in[17]. Embryos were obtained from natural crossings in mass-crossing cages (4 males + 4 females). The freshly laid eggs of MZ*nanog* mutants and wild-type were collected in 10-15 min intervals. The embryos were incubated at 28.5 °C and enzymatically dechorionated with 30 mg/ml Pronase shortly before 4.3 hpf. Embryos were homogenized in 10 ml 0.5 % Danieau's (for 1 L of 30X stock: 1740 mM NaCl, 21 mM KCl, 12 mM MgSO4, 18 mM Ca(NO3)2, 150 mM HEPES buffer, pH 7.6) containing 1x protease inhibitor cocktail (PIC, Roche) using a Dounce tissue grinder, and immediately treated with 1% (v/v) Methanol-free Formaldehyde (Pierce) for exactly 10 min shaking at room temperature. The fixation was stopped with 0.125 M Glycine by shaking for 5 min on a rotating platform. Lysate was centrifuged for 5 min, 4700 rpm at 4 °C, the pellet was resolved in cell lysis buffer (10 mM Tris-HCl (pH 7.5), 10 mM NaCl, 0.5 % NP-40, 1-4 ml/1000 embryos), followed by 5 min incubation on ice, and centrifuged 1 min, 2700 g at 4 °C. The pellet was washed 2 times with 1 ml ice cold 1x PBST (for 1 L: 40 ml PO4 buffer (0.5 M), 8 g NaCl, 0.2 g KCl, 0.1% Tween20, pH 7.5) and resuspended in 1 ml PBST. 5 μl aliquote of nuclei suspension was stained with Sytox green, nuclei were scored under fluorescence microscope using the Neubauer counting chamber. The residual nuclei were pelleted by 1 min centrifugation at 2700 g at 4 °C, snap frozen in liquid nitrogen and stored at −80 °C. The nuclei collected in different days were pooled together to reach the total number of 2.5-3 million, which was used to start one ChIP experiment.

The chromatin was thawn, pooled in 2 ml of nuclei lysis buffer (50 mM Tris-HCl (pH 7.5), 10 mM EDTA, 1 % SDS) and incubated 1 h on ice. In order to shear the chromatin to 200 bp fragments (on average), the chromatin was sonicated using the Covaris S2 sonicator (DC 20 %, Intensity 5, Cycles of Burst 200, Time = 3*40 cycles with 30 sec each (3*20 min)). To control the sonication, 30 μl of the sheared chromatin was de-crosslinked with 250 mM NaCl overnight at 65 °C and then analyzed using the Agilent Expert 2100 Bioanalyzer® and Agilent high sensitivity DNA Chip kit.

The lysed and sheared samples were centrifuged for 10 min, 4 °C at max. speed in table top centrifuge. 60 μl of each sample were kept as input control. The chromatin was then concentrated to 100 μl using the Microcon centrifugal filters (Merck Millipore MRCF0R030) and diluted 1:3 by adding ChIP dilution buffer (16.7 mM Tris-HCl pH 7.5, 167.0 mM NaCl, 1.2 mM EDTA) containing protease inhibitors. 3 μl of the H3K27ac antibody (anti-Histone H3 (acetyl K27) antibody - ChIP Grade (Abcam, ab4729, rabbit polyclonal, Lot: GR3357415-2, stock concentration 1 mg/ml, final dilution 10 ng/μl, validated for

zebrafish in[17]) was added and incubated overnight at 4 °C on a rotating wheel. 150 μl of magnetic Dynabeads coupled to protein G (Stock 30 mg/ml; invitrogen DynaI 10003D) were transferred into a 2 ml eppendorf tube and placed on a magnetic rack. The beads were washed 3x with 5 mg/ml specially purified BSA in PBST and 1x with 500 μl ChIP dilution buffer. After removing the ChIP dilution buffer, the chromatin-antibody mix was added and incubated with the beads at 4 °C overnight on a rotating wheel. Beads were pulled down by placing the eppendorf tubes on the magnetic rack. The beads were resuspended in 333 μl RIPA buffer (10 mM Tris-HCl (pH 7.6), 1 mM EDTA, 1 % NP-40, 0.7 % sodium deoxycholate, 0.5 M LiCl) containing PIC. The Protein G-antibody-chromatin complex was washed 4×5 min on a rotating platform with 1 ml of RIPA buffer, followed by 1×1 ml TBST buffer (25 mM Tris-HCl, 150 mM NaCl, 0.05% Tween 20, pH 7.6). The beads were pulled down again. To elute the chromatin, 260 μl elution buffer (0.1 M NaHCO3, 1% SDS) was added and incubated for 1 h at 65 °C in a water bath. The samples were vortexed every 10 – 15 min. The beads were pulled down, supernatant was transferred to a fresh eppendorf tube. 12.5 μl 5 M NaCl was added to de-crosslink the chromatin and incubated overnight at 65 °C in a water bath. The input samples were treated in parallel (230 μl elution buffer per 30 μl input). Purification of the de-crosslinked chromatin was performed using the QIAquick PCR Purification Kit from Qiagen. The concentration was determined using the Qubit fluorometer and Quanti-iT™ PicroGreen® dsDNA Kit according to manufacturer instructions.

To estimate the signal to background ratio in each ChIP experiment, we used the positive and negative reference genomic regions, near *tiparp* and *igsf2* genes, respectively enriched in or devoid of H3K27ac (Gao et al., 2022). We performed quantitative PCR in ChIP and Input control material, using the primers for these regions. PCR primers used were: tiparp_f_1 5' CGCTCCCAACTCCATGTATC-3', tiparp_r_1 5'-AACGCAAGCCAAACGATCTC-3', igsf2_f_2 5'-GAACTGCATTAGAGACCCAC-3', igsf2_r_2 5'-CAATCAACTGGGAAAGCATGA-3'. QPCR was carried out using the SsoAdvanced™ Universal SYBR® Green Supermix from BIO-RAD. ChIP and input were normalized by ddCT method, using negative reference region (*igsf2*). The ChIP experiment was considered successful, if the enrichment in ChIP over input control on the positive reference region (*tiparp*) was more than 5-fold.

The library preparation was carried out with NEBNext® Ultra™ DNA Library Prep Kit according to manufacturer instructions, with the modifications indicated below. As the DNA outcome of individual MZ*nanog* K27ac ChIP experiments did not reach the input DNA limit for the kit (5 ng), we pooled together the material from two successful ChIP experiments, as well as corresponding inputs. Since the DNA input was <100 ng, in the adaptor ligation step, the NEBNext Adaptor for Illumina® (15 μM) was diluted 10-fold in 10 mM Tris-HCl (pH 7.8) to a final concentration of 1.5 μM and used immediately. At the final clean-up step, the reaction was purified by mixing the samples with AMPure XP Beads (45 μl) before incubating at room temperature for 5 min. After a quick spin using the tabletop centrifuge, samples were placed on a magnetic rack and supernatant was discarded. 200 μl of 80 % Ethanol were added and removed after 30 sec, two times. The beads were air dried for 3 min and the DNA target was eluted by adding 33 μl of 0.1x TE (pH 8) and incubating at room temperature for 2 min. 28 μl of the library were transferred to a fresh PCR tube and stored at −20 °C. 2 μl of the sample were diluted 5-fold with 0.1x TE and used to check the size distribution of the library using Agilent Expert 2100 Bioanalyzer® and Agilent high sensitivity DNA Chip kit. The ChIP-seq library was sequenced at 70 mln paired end 150 bp reads the input library were sequenced to 30 mln reads. Sequencing was performed by the Novogene company (China). The raw and processed data have been deposited in GEO under the number GSE143439.

**Definition of regulated enhancers.** Sequenced reads were mapped to the zebrafish danrer11 assembly using Bowtie2 in the european Galaxy

server. Alternate loci and unassembled scaffolds were excluded. H3K27ac data for MZ*nanog* from this work was further processed together with H3K27ac for the wild-type, MZ*spg* and MZ*sox19b*[17]. To create bigwig files for scoring, the log2 ratio between each ChIP and merged inputs (in rpkm) was obtained using BAMcompare program in deepTools, bin size =10. The mean H3K27ac signal (log2 ChIP/input per bin) around each AR was calculated for 1 kb around AR summit. Values smaller than 0.1 were rounded to 0.1. We considered AR as H3K27-acetylated, if the mean H3K27ac signal was higher than arbitrary threshold 0.2. All H3K27-acetylated TdARs were considered as enhancers. Enhancer was considered as activated by TF, if log2 (mutant by TF/WT) ratio of H3K27ac signals was less than arbitrary threshold −0.7, was considered as blocked by TF, if log2 (mutant by TF/WT) ratio of H3K27ac signals was more than 0.7, and was considered as unchanged by TF all other cases. Enhancers activated, blocked or unchanged by the sum of TFs were assigned by applying the same procedure and thresholds to H3K27ac ChIP-seq data on the WT and MZ*nps*[5]. H3K27ac status of all ARs is provided in the Source Data file 1.

**Gel retardation assays (electrophoretic mobility shift assays, EMSA).** Pou5f3-3xFLAG[12] and Nanog-HA (kind gift of Nadine Vastenhouw) expression plasmids were in-vitro translated using TnT® Quick Coupled Transcription/Translation Systems (Promega) according to the instructions. Fifteen 20-28 bp motif-containing sequences (Fig. S6a) were selected from enhancers, based on ATAC-seq like the following: 1.P + N + : chromatin accessibility was reduced in MZ*spg* and MZ*nanog*; 2.P + N- accessibility reduced in MZ*spg* and increased in MZ*nanog*, 3.N + P- accessibility increased in MZ*spg* and reduced in MZ*nanog*. Forward and reverse EMSA oligos are listed in the Resources table, genomic positions of the enhancers in the Source Data file 1. Three G bases were added at the 5′ end of each EMSA oligo to enable radioactive labelling. Radioactively labeled probes were generated by annealing complementary EMSA oligonucleotides (45 μM, NEB2 buffer (NEB), heating to 95° for 2 min and cooling down to 25° with −1 °C per minute. Protruding 5′ ends were filled in with Klenow enzyme (NEB) in the presence of [P]−32 dCTP (Hartmann Analytics, SCP-405H). 10 μl labeling reactions containing 2 pM annealed oligo, 5 μl [P]−32 dCTP (5 μM), 1 μl dATP, dCTP and dGTP 2 mM mix, 1 μl NEB2 buffer and 0.1 U enzyme were incubated 15 min 30 °C and purified from unincorporated nucleotides using Nucleotide Removal Kit (Qiagen). Binding reactions were carried out in a total volume of 10 μl, containing 2,5 μl 4x binding buffer (40% glycerin, 80 mM HEPES (pH 7.6); 4 mM DTT; 0.04% NP-40 supplement (Roche), 4x Complete protease inhibitors (Roche), 5 mM MgCL2, 600 mM KCl), 500 ng poly dIdC (non-specific inhibitor), 1 or 2 μl of in-vitro-translated proteins, and 20 femtomoles of radioactively-labeled probe ( >10 000 cpm). In the experiment shown at Fig. S6a, additionally 50 ng of salmon sperm were added as non-specific inhibitor. 15 ng of specific inhibitor (unlabeled double-stranded oligo) or 1 μl of anti-FLAG (clone M2 Sigma F3165, lot number SLCJ3741, stock 10 mM, final concentration 1 mM, validated in[12]) or anti-HA (Roche/Sigma #1186743001, Rat anti-HA High-affinity monoclonal, Clone 3F10, Lot 1564900, stock 100 μg/ml, was used in final concentration 10 μg/ml) antibodies were added to control the reaction specificity. After incubation for 30 min at 4 °C, the reactions were analyzed by non-denaturing 5% polyacrylamide gel electrophoresis, followed by drying the gels and autoradiography. Non-denaturing 5% polyacrylamide 1xTBE Criterion gels (BioRad) were pre-run 30 minutes before loading the reactions.

**RNA-seq time curves.** RNA-seq time curves were obtained as described previously (Gao et al., 2022). Freshly laid eggs were obtained from natural crossings in mass-crossing cages (4 males + 4 females). 5-10 cages were set up per genotype, the eggs from different cages were pooled. At least 600 freshly laid eggs per each genotype collected within 10-15 minutes were taken for single experiment. To match the

developmental curves as precisely as possible, the material was simultaneously collected for two genotypes in parallel. 45-60 minutes after the egg collection, we ensured that the embryos of both genotypes are at 2-4 cell stage, removed non-fertilized eggs and distributed the embryos to 8 dishes per genotype, 40-45 embryos per dish, to obtain 8 time points per genotype: 2.5, 3, 3.5, 4, 4.5, 5, 5.5 and 6 hpf. The temperature was kept at 28.5 °C throughout the experiment. The embryos from one dish per genotype were snap-frozen in liquid nitrogen every 30 minutes, starting from 2.5 hpf (pre-ZGA, 256 cell stage, 8th cell cycle) till midgastrula (6 hpf). Two biological replicates for each of MZ*nanog* and MZ*sn* were previously collected in parallel to the wild-type replicates B1 and B2 from (Gao et al., 2022). In addition, we collected six biological replicates of the time curve for the wild-type, and three biological replicates for MZ*pn* and MZ*triple* mutants from six independent experiments. Total RNA was isolated with QIA-GEN RNeasy kit following the user's manual. We checked RNA quantity and quality by using Agilent RNA 6000 nano Kit on Agilent Bioanalyzer, according to manufacturer instructions. Poly-A enriched library preparation and sequencing on Illumina platform was performed by Novogene company (China). The MZ*nanog* and MZ*sn* sampes were single-end sequenced at 35 million reads per sample (50 bp), similarly to the previously published samples WT, MZ*sox19b*, MZ*spg,* and MZ*ps*[17]. The MZ*pn* and MZ*triple* samples and corresponding six replicates of the wild-type were paired-end sequenced at 58 million reads per sample (150 bp).

**RNA-seq time curves: data processing.** FASTQ files were processed on european Galaxy server usegalaxy.eu. All sequenced clean reads data were trimmed on 5′ end for 5 bp by Trim Galore! program according to the sequencing quality. Trimmed reads were mapped to danrer11 genome assembly using RNA STAR[45] and ENSEMBL gene models. Number of reads for each gene was counted using Feature count[46]. Single-end and paired-end sequencing data sets were further processed separately as the part 1 and the part 2. For the part 1, feature counts for the MZ*nanog* and MZ*sn* samples were cross-normalized with MZ*sox19b*, MZ*spg* and MZ*ps* and four biological replicates of the wild-type[17] using Deseq2. For the part 2, feature counts for the MZ*pn* and MZ*triple* samples and corresponding six replicates of the wild-type were cross-normalized using Deseq2. The raw data and two normalized data tables were deposited in GEO with the accession number GSE162415.

**Finding differentially expressed zygotic genes.** We used 3-step RNA-sense program to process normalized RNA-seq time curves as described in[17]. Shortly, RNA-sense takes RNA-seq time curves in two conditions (WT and mutant) as an input. The transcripts expressed below a user-defined threshold are excluded from the analysis. In the first step, RNA-sense creates the lists of transcripts switching UP or DOWN in time, in one of the genotypes; the genotype and switch p-value are defined by user. We found that 4247 genes are switching UP in the wild-type, in both part1 and part 2 of the analysis (thresholds >100 for expression and <0.15 for switch p-value). Switching UP transcripts were considered as zygotically expressed[17]. 1668 transcripts were switching up in at least one mutant but not in the wild-type; we considered them as zygotically expressed in the respective mutants. The regulation status was assigned to each zygotic transcript on the second step of RNA-sense. Zygotic transcript was considered to be downregulated in the mutant, if it was expressed at least two-fold lower compared to the wild-type with p-value < 0.05 in Student's t-test in at least one time point at or after the switch UP. Zygotic transcript was considered to be upregulated in the mutant, if it was expressed at least two-fold higher compared to the wild-type with p-value < 0.05 in Student's t-test in at least one time point. Some transcripts were up – and down regulated in the same mutant in different time points and were considered as "undefined" ("U"). The list of zygotic genes, their switch and regulation status in all the mutants is provided in the Source Data file 6.

**Enrichment statistics.** Zygotic transcripts (Source Data file 2) were linked to Chromatin Accessible Regions (ARs, Source Data file 1) within +/−50 kb from Transcription Start Site (TSS). Resulting 70419 AR-transcript pairs were uniquely numbered (Source Data file 3). To estimate the over- or underrepresentation of different types of enhancers in the putative regulatory regions of zygotic genes, Chi-squared ($\chi^2$) test was performed using chisq.test function and visualized using corrplot function in R. We considered only strong positive correlations with Pearson residuals more than 4. To reproduce our results, the input files for all $\chi^2$ tests can be easily derived from the Source Data file 3 as explained below. The input file is a tab-separated text file with three columns: "AR-transcript_pair" (which is always column A in the Source Data file 3), "AR", and "transcript". To create the input file for $\chi^2$ test, delete the first two rows of the Source Data file 3, keep column A and two columns specified below (delete all other columns), delete the rows specified below, change the headings to AR" and "transcript" and export the input file as tab-separated text. Figure 2g: "AR" – column N, "transcript" – column AF, delete all rows with "U". Figure 5d-f: "AR" – columns O, Q, R, respectively, delete all rows with "none"; "transcript" – column AG, delete all rows with "N/A". Figure 6c: "AR" – columns G, Q, R, delete all rows with "none"; "transcript" – column Z, delete all rows with "U". Figure 6c: "AR" – columns G, Q, R, delete all rows with "none"; "transcript" – column AB, delete all rows with "U". Fig. S10b, c: "AR" – columns G, P, respectively, delete all rows with "none"; "transcript" – column AG, delete all rows with "N/A". Fig. S10f: "AR" – columns G, Q, R, delete all rows with "none"; "transcript" – column AC, delete all rows with "U".

**Gene ontology analysis.** For the genomic regions we used GREAT analysis[47]. Genomic regions were converted to zv9/danrer9 genomic assembly, which is the only assembly available for this server, using Liftover Utility from UCSC and associated with genes using 20 kb single nearest gene association rule. The categories were ranked by ascending FDR value. For the transcript groups we used DAVID[48].

**Data visualization.** Omni-ATAC-seq: to create bigwig files for data visualization, Omni-ATAC-seq signal (in rpkm) was obtained using BAMcoverage program in deepTools2, bin size =10. H3K27ac: to create bigwig files for data visualization, the log2 ratio between each ChIP and merged inputs (in rpkm) was obtained using BAMcompare program in DeepTools, bin size =10. The heatmaps or mean profiles were plotted using plotheatmap or plotprofile programs in DeepTools. H3K27ac, TF ChIP-seq and omni-ATAC-seq profiles in single genes were visualized in UCSC browser. RNA-seq heatmaps were plotted using R with custom script. For the heatmaps, the transcripts were sorted by ascending RNA-sense switch p-value then by ascending RNA-sense switch time then by group. Biological replicates from all experiments were cross-normalized using Deseq2. Replicates for each genotype were averaged and the maximal expression value across the time curve in all genotypes was calculated. Expression/max ratio for each timepoint was plotted.

**Mathematical modeling.** The mathematical modeling process was performed in two main steps, (i) the core model and (ii) the mini models. The idea behind developing a core model was to use the resulting dynamics of relevant transcription factors or their combinations as input functions to the mini models that later describe the dynamical behavior of a huge amount of 1799 preselected target genes. Generation of ODE models, parameter estimation and identifiability analysis was performed by means of the R package dMod[49]. For each model, the best parameter set was obtained by the method of maximum likelihood performing a deterministic multi-start optimization[50] from multiple randomly chosen starting points in parameter space. The trust region optimizer *trust* (Geyer CJ (2004) trust: Trust Region Optimization. R package Version 0.1-8) was used constituting the standard optimizer in the R package dMod. Model parameters were

log-transformed to ensure positivity and enable optimization over a broad range of magnitudes[50]. Parameter values were not restricted by fixed borders. Instead, in order to prevent the optimizer from finding solutions with very low or high parameter values, we constrained the model parameters with a weak $L_2$ prior that contributed with one to the likelihood, if the parameter differed by three orders of magnitude from a value of zero on the logarithmic scale. For the computation of confidence intervals and to test identifiability of the model's parameters by means of the profile likelihood[51] the contribution of these $L_2$ priors was subtracted to ensure a purely data-based result.

**Construction of the core model.** In total, the core model consists of 11 model states and 17 reactions (Fig. S7a, Source Data file 5). Kinetic expressions of the model reactions were derived based on state-of-the-art transcription factor network modeling[52] and prior knowledge from published literature[14,17,19,20]. For simplicity, the degradation of Sox19b was incorporated via a switch-like input function dependent on the time t as a parameter. Model states were assumed to be directly observed, and uncertainty of the observations were estimated jointly with the kinetic parameters assuming an absolute error model for the measurements of each observed state. Model equations, as summarized in Source Data file 5, include three boolean variables MZ*spg*, MZ*nanog* and MZ*sox* that were used to distinguish between the seven experimental conditions including wild type, single, double and triple knockdowns. In total, a number of 888 RNA-seq data points for six targets was used for model calibration. The first time point of the data set was at 2.5 hpf. As an assumption, the initial values of the ODE model were defined at 2 hpf. The initial values of downstream targets Sox2, Sox3, and Sox19a were assumed to be zero, while for the values of the zygotic transcription factors Pou5f3, Sox19b and Nanog a free parameter was estimated by the model, with the constraint of being the same overall experimental conditions.

During the model development, a couple of parameters was fixed to zero or one, following likelihood-ratio test-based model reduction[53] since they were not necessary in order to describe the available data. Hill-kinetic parameters were fixed to a value of five corresponding to a strong threshold behavior. Further parameter transformations that have been applied are summarized in[54]. Parameter estimation was performed using deterministic multi-start optimization[50] starting from 192 randomly chosen sampled positions in parameter space. Parameter values of the best obtained fit are summarized in Source Data file 6. To show reliability of the optimization, the 100 best optimization runs were displayed as a waterfall plot (Fig. S8b) sorted by their negative log-likelihood values[50]. The global optimum was found in 38 of the 101 converged optimizations runs. Considering a confidence level of 95%, profile likelihoods showed finite confidence intervals for 30 out of 37 parameters (see Fig. S9a).

**Construction of the mini models.** Assuming at least one of Pou5f3, Nanog and SOX to be activating, a set of 19 mini models was constructed. Model equations for each mini model (Source Data file 7) were derived from the theory of transcription factor network modeling[52]. When more than one transcription factor was either activating or repressing, the combination was modeled using appropriate logical AND-gate functions[53]. Dynamics of the proteins Pou5f3, Nanog, and the sum of SoxB1 class genes SOX were taken from the core model for the different experimental conditions and used as input functions for the mini models. To avoid numerical issues with infinite derivatives of hill-kinetic terms, a small number of 1e-4 was added to the input functions where necessary.

Analogously to the core model, RNA-seq data of seven experimental conditions was used for model fitting of each mini model and for each target gene. Per target gene, a total number of 192 was available. In contrast to the core model, the different experimental conditions were not incorporated via boolean variables, instead, the

differential equations stayed the same over all conditions. Thus, differences in dynamics were only possible via the differences in the given input dynamics of the core model.

For each mini model and for each target gene, a total number of 64 optimization runs was performed out of which 99.97% converged and at least 98.67% reached the global optimum. Finally, using the Bayesian Information criterion (BIC), out of the 19 analyzed mini models, the best fitting one was selected for every target. Parameter values of the selected models are summarized in Source Data file 8.

### Quantification and statistical analysis

Statistical comparisons of three or more samples were performed using one-way ANOVA, followed by Tukey-Kramer Test to evaluate the statistical significance of pairwise differences. Two samples were compared using Student 2-tailed t-test.

### Reporting summary

Further information on research design is available in the Nature Portfolio Reporting Summary linked to this article.

## Data availability

The raw and processed ATAC-seq data generated in this study have been deposited in the GEO database under accession code GSE215956. The processed ATAC-seq data generated in this study are provided in the Source Data file 1. The raw and processed RNA-seq data generated in this study have been deposited in the GEO database under accession code GSE162415. The processed RNA-seq time curves data generated in this study are provided in the Source Data files 2. The linked ATAC-seq and RNA-seq data generated in this study are provided in the Source Data file 3. The ChIP-seq data generated in this study have been deposited in the GEO database under accession code GSE143439. The processed ChIP-seq data generated in this study are provided in the Source Data file 1. The uncropped EMSA images generated in this study are provided in the Source Data file 4 and at the end of the Supplementary Information file. The model equations generated in this study are provided as Source Data file 5 and Source Data file 7. The best fit parameters for the models generated in this study are provided as Source Data file 6 and Source Data file 8. Source data are provided with this paper.

## Code availability

The original code is deposited in Github [https://github.com/vandensich/zebrafish-minimodels][54].

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

## Acknowledgements

We are grateful to Nadine Vastenhouw and Giorgos Pyrowolakis for providing the plasmids and reagents, to Rainer Duden for commenting on the manuscript, to Sabine Götter for fish care, and to Andrea Buderer and Cornelia Wagner for administrative support. This work was supported by DFG-ON86/4-2 and DFG-ON86/6-1 for DO, and DFG-EXC2189 – Project ID: 390939984 for D.O. and J.T. BG is funded by DFG grant SFB 992/1 2012 and BMBF grant 031 A538A RBC.

## Author contributions

A.J.R., M.G., M.V., D.O., and A.G. - experiments; A.J.R., M.G., B.G., and D.O. - data analysis, M.R., H.H., and J.H. -mathematical modeling, D.O., J.T. - supervision of the study. D.O. wrote the manuscript, all authors edited the manuscript.

## Funding

## Competing interests

The authors declare no competing interests.
