## [Peer Review File · Nature Communications]

Activator-blocker model of transcriptional regulation by pioneer-like factorsREVIEWER COMMENTS

Reviewer #1 (Remarks to the Author):

In this manuscript, Riesle and Gao et al. analyzed ATAC-seq, H3K27ac ChIP-seq, and transcriptome of zebrafish mutants of ZGA regulator TFs. They found the existence of synergistic and antagonistic enhancer types, and Pou5f3 and Nanog can function as activator or blocker in antagonistic enhancers. The topic is unquestionably important for the field, but this reviewer feels that the novelty of the study seems weak. I have several major concerns as described below.

Regarding the novelty, the fact that the pioneer factors (Pou5f3 and Sox19b) is preventing premature expression for some of the genes during ZGA has already been reported by the same group (Gao et al., 2022). The idea that Pou5f3 and Nanog is competing at a same binding site and may function as a blocker is indeed interesting and advancing our knowledge, but molecular experimental evidence is lacking. How Pou5f3 and Nanog function as a non-pioneer blocker is also completely a blackbox. To address underlying molecular mechanisms, at least the authors should do experiments such as gel-shift assay to enhancer sequences of specific group of genes.

The authors claim the presence of additional genome activators, but the evidence is not enough. First, in Fig. 2a, why did not the authors use the MZsox19b for the classification? Second, the results show that ~50% or ~80% of group 4.- TdARs can be rescued by Sox19b alone, or by Sox19b and Nanog, respectively (Fig S3). From these data, it is likely that SoxB, or SoxB and Nanog is required on chromatin accessibility of most of group 4.- TdARs.

Characterization of antagonistic enhancers is not enough. They show GC content for 1.PN, 2.P, 3.N, and 4.-, but comparison between p-n+, p+n- and others is required. Furthermore, they mention that exact match to the motif is important for Pou5f3 and Nanog to function as pioneer TFs, but this does not explain the difference between synergistic and antagonistic enhancers. Why do Pou5f3 and Nanog function as blocker only in antagonistic enhancers, but not in synergistic enhancers? The author also need to have additional experimental data as stated above.

Regarding the mathematical modelling; in Figure 5c, expressions of 3b. P-N+ genes do not seem to be upregulated in MZps. They even seems down regulated at stages before 6 hpf. Furthermore, gene ontology analyses are not enough to evaluate the model accuracy, and validation in other method is required. For example, is there any differences in sequence feature between enhancers of the six groups sorted by the model?

Reviewer #2 (Remarks to the Author):

The manuscript "Activator-blocker model of transcriptional regulation by pioneer-like factors," by Riesle et al takes a deep dive into functions for the transcription factors Pou5f3, Sox19b and Nanog during zebrafish zygotic genome activation. Their analysis uses a combination of elegant genetic and genomic approaches. The results reveal surprising complexities in the ways in which Pou5f3, Sox19b and Nanog work both together and antagonistically to regulate early transcription.

Overall this is a rigorous study which produces useful data sets and shifts the way we need to think about how these transcription factors act during zebrafish ZGA. I have only minor comments.

The authors do a very nice job of detailing the crosses performed to generate the various mutant lines in the methods, and the original alleles are cited. However, given the importance of the mutants it might be worth reviewing the nature of the alleles in the start of the results section. For example, do any of them produce proteins that could still have some DNA binding capacity?

Figure 2b, not clear exactly what control genomic regions are.

With respect to figure 2, it could be more helpful to include a little more detail on the binning of up down and unchanged groups in the text or figure legend (what were the cutoffs/criteria for including). I similarly struggled to find this information in the methods, if it is there it is not easy to find.

Figure 3 panel 1 is a little confusing. Is purple color code also needed?

With respect to figure 3 F, it isn't entirely clear to me why this group couldn't reflect a requirement for all three TFs together, rather than any one plus GC binding factor- Fig 3b seems to suggest binding motifs for all three are detected in this group, although the sox:pou motif is underrepresented. The GC enrichment could still represent a hypothetical GC binding factor required in addition to the three factors? Alternatively, is it possible that there is simply a structural property of these GC rich regions that is keeping them more open? I believe there is reasonable data to support a relationship between GC content and nucleosome occupancy

The rescue experiments in Fig S3 are quite nice. I almost wonder if they might be better in the main text of the manuscript.

Figure 4, Panel C in particular is really important and very difficult to interpret, not at all intuitive. Is there any other way to present this data more clearly? I had to sit with the browser shots in panel H a good while to understand what was going on holistically in figure 4.

I think a prediction of the model in 4f is that you should never be able to IP A and B from chromatin together at these types of sites. Is it possible to do sequential IPs to explore that possibility? This could be interesting additional support, but if this is technically too challenging, could be viewed as beyond the current paper's scope.

Dear reviewers #1 and #2,

We are grateful that you professionally spotted the weak points in our manuscript, which helped us to improve it. We performed additional experiments, added new sub-chapter in the results section with new main Fig. 5 and Fig.S6, performed additional analysis and made changes in most of the figures. We numbered your comments and linked them as comments to the yellow-marked text in the PDF file "Riesle_main_and SUPPL_marked for reviewers", which contains main text and figures, supplementary figures and legends and also additional supplementary Figure for Reviewer 1, at the end of the supplementary material. Please find detailed answers to your criticisms below. Sincerely,
the authors.

REVIEWER COMMENTS

Reviewer #1 (Remarks to the Author):

In this manuscript, Riesle and Gao et al. analyzed ATAC-seq, H3K27ac ChIP-seq, and transcriptome of zebrafish mutants of ZGA regulator TFs. They found the existence of synergistic and antagonistic enhancer types, and Pou5f3 and Nanog can function as activator or blocker in antagonistic enhancers. The topic is unquestionably important for the field, but this reviewer feels that the novelty of the study seems weak. I have several major concerns as described below.

R1-1.Regarding the novelty, the fact that the pioneer factors (Pou5f3 and Sox19b) is preventing premature expression for some of the genes during ZGA has already been reported by the same group (Gao et al., 2022). The idea that Pou5f3 and Nanog is competing at a same binding site and may function as a blocker is indeed interesting and advancing our knowledge, but molecular experimental evidence is lacking. How Pou5f3 and Nanog function as a non-pioneer blocker is also completely a blackbox. To address underlying molecular mechanisms, at least the authors should do experiments such as gel-shift assay to enhancer sequences of specific group of genes.

A 1-1. We agree with the comment and provide requested evidence. We performed series of the gel-shift assays with the oligos from different enhancers, added the new figures 5 and S6 and a chapter "**Pou5f3 and Nanog bind to the common binding sites in a mutually exclusive way**". We also show that in most cases TdARs have only one match to either Pou5f3 or Nanog motif (new panel Fig. S5f).

R1-2. The authors claim the presence of additional genome activators, but the evidence is not enough. First, in Fig. 2a, why did not the authors use the MZsox19b for the classification? Second, the results show that ~50% or ~80% of group 4.- TdARs can be rescued by Sox19b alone, or by Sox19b and Nanog, respectively (Fig S3). From these data, it is likely that SoxB, or SoxB and Nanog is required on chromatin accessibility of most of group 4.- TdARs.

A1-2. We agree with both points. In response to this criticism we
- made additional analysis of 4.- group, using ATAC-seq in MZsox19b and three double mutants (new panel c in the Fig. S3).
- removed "hypothetical GC protein" from the scheme in Fig.3; and speculations about "hypothetical GC protein" from the corresponding text in the results.

R1-3. Characterization of antagonistic enhancers is not enough. They show GC content for 1.PN, 2.P, 3.N, and 4.-, but comparison between p-n+, p+n- and others is required.

A1-3. Done (Fig. S4). Conclusion: overall GC content of enhancers activated by Pou5f3 is lower than of the enhancers activated by Nanog. This is seen in both ATAC-seq (Fig.3e, Fig. S4b) and H3K27ac comparisons between TdARs (Fig. S4b).

R1-4. Furthermore, they mention that exact match to the motif is important for Pou5f3 and Nanog to function as pioneer TFs, but this does not explain the difference between synergistic and antagonistic enhancers. Why do Pou5f3 and Nanog function as blocker only in antagonistic enhancers, but not in synergistic enhancers? The author also need to have additional experimental data as stated above.

A1-4. There are two parts of the answer to this criticism:

- 1) Yes, we provided additional experimental data and confirmed now with gel-shifts, that the factor which binds stronger in-vitro works as an activator in-vivo (9 out of 9 oligos with single motif, where the binding worked for any of the two TFs, see the new Fig. 5 panel e). We also show that GC content is important: Pou5f3 activates the enhancers with lower GC content range, than Nanog, as judged by pioneer activity (Fig.3e) and H3K27ac change (Fig.S4).

- 2) No, we can not explain the mechanistic difference between synergistic and antagonistic enhancers from gel-shifts and bulk genomic data. We assume that cell-specific cofactors are involved (see discussion). Single-cell analysis is required to answer this question; as far as we know ChIP-seq technique for single cells is not yet developed. We hope that our manuscript is novel enough to be published without it..

R1-5. Regarding the mathematical modelling; in Figure 5c, expressions of 3b. P-N+ genes do not seem to be upregulated in MZps. They even seems down regulated at stages before 6 hpf.f

This is because most of 3b. P-N+ genes are coactivated by SOXB1 sum, and SOXB1 sum is decreased in MZps.(see the supplementary Figure for Reviewer 1 included in the file for reviewers). 3b. P-N+ model group consists of three mini-models: of three mini-models: S+P-N+ (best fit to 398 transcripts), S0P-N+ (best fit to 24 transcripts) and S-P-N+ (best fit to 6 transcripts).

In response to this criticism we changed the sumentary figure S8, which shows now the example fits to all mini-models (and not to the groups as before).

R1-6. Furthermore, gene ontology analyses are not enough to evaluate the model accuracy, and validation in other method is required.

We did not intent to validate the modeling with GO analysis. In response to this criticism we completely removed this GO analysis from the paper, not to distract the reader attention. We renamed the sub-chapters and put the cross- validation part (the synergistically and antagonistically regulated transcripts are linked to synergistically and antagonistically regulated enhancers) just after the description of the modeling results (see the marking of the text for reviewers).

R1-7.For example, is there any differences in sequence feature between enhancers of the six groups sorted by the model?

Yes, there is a difference in both sequence features: motif frequency and in GC content. We show it now in Fig. S9 c,d, as additional validation.

Reviewer #2 (Remarks to the Author):

The manuscript "Activator-blocker model of transcriptional regulation by pioneer-like factors," by Riesle et al takes a deep dive into functions for the transcription factors Pou5f3, Sox19b and Nanog during zebrafish zygotic genome activation. Their analysis uses a combination of elegant genetic and genomic approaches. The results reveal surprising complexities in the ways in which Pou5f3, Sox19b and Nanog work both together and antagonistically to regulate early transcription.

Overall this is a rigorous study which produces useful data sets and shifts the way we need to think about how these transcription factors act during zebrafish ZGA. I have only minor comments.

R2-1. The authors do a very nice job of detailing the crosses performed to generate the various mutant lines in the methods, and the original alleles are cited. However, given the importance of the mutants it might be worth reviewing the nature of the alleles in the start of the results section. For example, do any of them produce proteins that could still have some DNA binding capacity?

A2-1. All the mutants are genetic nulls: Pou5f3 mutant has a point-mutation before DNA-binding domain, Sox19b and Nanog have TALEN- induced frameshifts before the DNA-binding domains. We mention it in the results and give the references to the original publications where it was characterized after each mutant name.

R2-2. Figure 2b, not clear exactly what control genomic regions are.

A2-2. Included in the legend for Figure 2b: "To obtain control genomic regions (co, dotted line), genomic coordinates of all ARs were shifted 1 kb downstream".

R2-3. With respect to figure 2, it could be more helpful to include a little more detail on the binning of up down and unchanged groups in the text or figure legend (what were the cutoffs/criteria for including). I similarly struggled to find this information in the methods, if it is there it is not easy to find.

A2-3. We agree (it is only in “Methods” and in too detailed). We included in the figure legend for Fig. 2a: “Three groups of accessible regions (ARs) were selected as follows: in “down” and “up” regions, ATAC-signal was reduced or increased, respectively, in six MZ *triple* biological replicates compared to seven wild-type biological replicates with false discovery rate < 5%. “same” – the remaining ARs which were considered unchanged”.

R2-4. *Figure 3 panel 1 is a little confusing. Is purple color code also needed?*

A2-4. Yes! We included it to Fig 3a and to Fig S3a.

R2-5. *With respect to figure 3 F, it isn't entirely clear to me why this group couldn't reflect a requirement for all three TFs together, rather than any one plus GC binding factor- Fig 3b seems to suggest binding motifs for all three are detected in this group, although the sox:pou motif is underrepresented. The GC enrichment could still represent a hypothetical GC binding factor required in addition to the three factors? Alternatively, is it possible that there is simply a structural property of these GC rich regions that is keeping them more open? I believe there is reasonable data to support a relationship between GC content and nucleosome occupancy*

A2-5. We agree, we do not really need the GC factor in the scheme. We replaced it, removed the speculations about GC protein from the text, and included additional analysis of the 4.- group to the Figure S3c.

R2-6. *The rescue experiments in Fig S3 are quite nice. I almost wonder if they might be better in the main text of the manuscript.*

A2-6. We agree. We moved the experiments from the Fig.S3 to Fig. 3, panels f,g.

R2-7. *Figure 4, Panel C in particular is really important and very difficult to interpret, not at all intuitive. Is there any other way to present this data more clearly? I had to sit with the browser shots in panel H a good while to understand what was going on holistically in figure 4.*

A2-7. We tried our best to make the things simpler to precept. As a result, we expanded the analysis, the panel C is now moved from Fig. 4 to the new Supplementary Fig.4 which is completely filled with analysis. The order of remaining panels in the main Fig. 4 is reorganized.

R2-8. *I think a prediction of the model in 4f is that you should never be able to IP A and B from chromatin together at these types of sites. Is it possible to do sequential IPs to explore that possibility? This could be interesting additional support, but if this is technically too challenging, could be viewed as beyond the current paper's scope.*

A2-8. Thank you for 1) the clear formulation of this very important message, and 2) for putting this request in the minor comments. We show now in the new panel of the supplementary Fig. 5f that most of the open regions bound by TFs have only one site, either for Pou5f3 or for Nanog. We have also done a set of gel retardation assays with tagged Pou5f3 and Nanog and supershifts, showing that Pou5f3 and Nanog do not bind together to the same DNA motifs in-vitro. We included the new Fig 5, Fig. S5 and new sub-chapter of the results: **“Pou5f3 and Nanog bind to the common binding sites in a mutually exclusive way”**.

REVIEWER COMMENTS

Reviewer #1 (Remarks to the Author):

The reviewer appreciates the authors' efforts in incorporating new data into the revised manuscript. However, I still have a concern. The authors have made an attempt to demonstrate the mutually exclusive binding of Pou5f3 and Nanog using gel-shift assays (Figure 5). Nevertheless, the data only indicate that either Pou5f3 or Nanog strongly binds, while the other protein binds weakly to the same oligo. It does not clearly show the blocking of one protein by the other. It would be valuable for the authors to provide evidence by demonstrating the actual replacement of Pou5f3 binding with Nanog binding as the concentration of Nanog increases and vice versa, using the gel-shift assay the authors established. The reviewer is sure that the authors would easily perform this kind of experiments. Given that the activator-blocker model is one of the most crucial assertions of this paper, the reviewer thinks, it is essential to experimentally validate the molecular mechanism.

Reviewer #2 (Remarks to the Author):

All of my concerns regarding this manuscript have been addressed.

REVIEWER COMMENTS

Reviewer #1 (Remarks to the Author):

R1-1

The reviewer appreciates the authors' efforts in incorporating new data into the revised manuscript. However, I still have a concern. The authors have made an attempt to demonstrate the mutually exclusive binding of Pou5f3 and Nanog using gel-shift assays (Figure 5). Nevertheless, the data only indicate that either Pou5f3 or Nanog strongly binds, while the other protein binds weakly to the same oligo. It does not clearly show the blocking of one protein by the other. It would be valuable for the authors to provide evidence by demonstrating the actual replacement of Pou5f3 binding with Nanog binding as the concentration of Nanog increases and vice versa, using the gel-shift assay the authors established. The reviewer is sure that the authors would easily perform this kind of experiments. Given that the activator-blocker model is one of the most crucial assertions of this paper, the reviewer thinks, it is essential to experimentally validate the molecular mechanism.

Answer:

I performed the requested EMSA experiments, included them as a new Supplementary Fig.7 and referred them in the text (please see the yellow marking in the PDF file for reviewers).

Sincerely,
Dr.Daria Onichtchouk.

Reviewer #2 (Remarks to the Author):

All of my concerns regarding this manuscript have been addressed.